# COMMA: A COMMUNICATIVE MULTIMODAL MULTI-AGENT BENCHMARK

## ABSTRACT

The rapid advances of multi-modal agents built on large foundation models have largely overlooked their potential for language-based communication between agents in collaborative tasks. This oversight presents a critical gap in understanding their effectiveness in real-world deployments, particularly when communicating with humans. Existing agentic benchmarks fail to address key aspects of inter-agent communication and collaboration, particularly in scenarios where agents have unequal access to information and must work together to achieve tasks beyond the scope of individual capabilities. To fill this gap, we introduce a novel benchmark designed to evaluate the collaborative performance of multimodal multi-agent systems through language communication. Our benchmark features a variety of scenarios, providing a comprehensive evaluation across four key categories of agentic capability in a communicative collaboration setting. By testing both agent-agent and agent-human collaborations using open-source and closed-source models, our findings reveal surprising weaknesses in state-of-the-art models, including proprietary models like GPT-4o. These models struggle to outperform even a simple random agent baseline in agent-agent collaboration and only surpass the random baseline when a human is involved.[1]

## 1 INTRODUCTION

The field of multimodal agents is experiencing rapid growth (Xu et al., 2024; Xie et al., 2024; Cao et al., 2024), with research efforts expanding at an unprecedented pace. However, amidst this growth, a critical gap in research has emerged: the lack of focus on collaborative work (Gurcan, 2024; Park et al., 2023; Hong et al., 2024; Liu et al., 2024) among multiple multimodal agents. Synergistic operation of such agents is a highly promising but largely unexplored domain. Language agents can collaboratively finish complex tasks such as software development by assuming functional roles such as system designer, function generator, etc (Qian et al., 2024; Du et al., 2024)). Current research on multimodal agents (Xu et al., 2024; Xie et al., 2024; Cao et al., 2024) has predominantly focused on individual agent capabilities, neglecting the potential for inter-agent collaboration. This limitation is further compounded by existing benchmarks such as VisualWebArena (Koh et al., 2024) and MME-RealWorld (Zhang et al., 2024), which fail to adequately assess collaborative performance between agents. As a result, our ability to evaluate and improve multi-agent systems remains constrained, hindering progress in this crucial area.

Several critical questions emerge in the context of multimodal agent collaboration. How can different agents effectively communicate multimodal information through language when they have varying levels of access to information? In scenarios where different agents possess diverse task-specific capabilities, how can they collaborate to accomplish objectives that are beyond the scope of any individual agent? These research settings remain largely uncharted and present significant challenges. Furthermore, the ability of agents to handle incomplete information is of paramount importance, particularly when working with sensitive data (Li et al., 2024) (i.e. Agent application in healthcare where privacy concerns are critical (Tang et al., 2024)). The exploration of these questions is crucial for advancing the field of multimodal agent collaboration. By addressing these challenges, we can expand the applicability of multimodal agents in real-world scenarios (Zhang et al., 2024), particularly those involving sensitive or restricted information.

---

[1]We will release our benchmark and evaluation code upon acceptance.

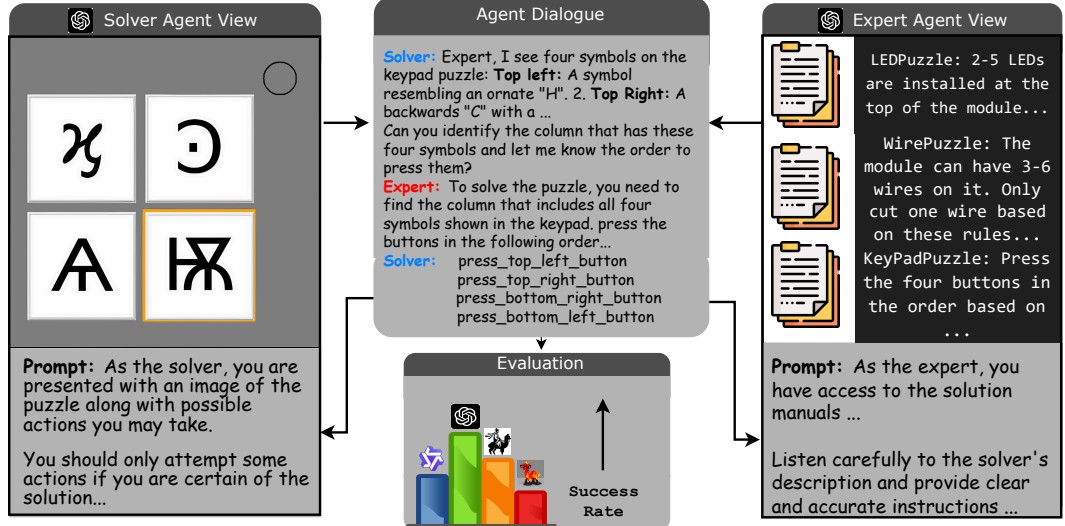

Figure 1: Overview of the interaction between the Solver and Expert agents in our benchmark. The game manager presents the **Solver** agent with a puzzle, where the **Solver** can choose to interact by clicking or requesting advice from the **Expert** agent. The **Solver** is shown an image of the puzzle (a KeyPad puzzle in this instance) and makes decisions based on the possible actions. The **Expert**, informed by instruction manuals, provides guidance based on the **Solver**'s descriptions, such as advising which buttons to press. The interaction between the **Solver** and **Expert** is captured in the dialogue, reflecting the cooperation necessary to complete the task. Both agents use self-reflection on their choices by being prompted with the conversation history as it progresses.

Motivated by these aforementioned issues, we propose a novel benchmark for evaluating collaborative multi-modal multi-agent frameworks to address critical gaps in current approaches (see Figure 1). Our evaluation setting simulates a scenario where an in-house agent with direct access to sensitive data (i.e., the AI solver) collaborates with external expert agents (i.e., the AI expert) to analyze information without compromising privacy. This evaluation setting could revolutionize how we handle and extract insights from sensitive datasets across various domains.

We assess multi-modal multi-agent systems using a series of carefully designed collaborative puzzle games. These scenarios typically involve two-player setups where agents have access to different, complementary information. (i.e., in a bomb defusal game, one agent possesses details about the bomb, while the other has access to a disarming manual). By employing such diverse and interactive scenarios, we aim to provide a thorough assessment of multi-modal multi-agent performance.

Our benchmark includes 10 distinct, easily customizable puzzles with thousands of unique solutions. We tested two different settings (AI-AI and AI-Human) and evaluated several popular multimodal models, including closed-source models (GPT-4V, GPT-4O, and GPT-4o1) and open-source models (Qwen-VL (Bai et al., 2023), and InternVL(Chen et al., 2024)). Surprisingly, the GPT series does not outperform even a simple random baseline in the AI-AI setting, highlighting a potential growth area for future model development. Our contributions are as follows:

- We propose an evaluation framework called COMMA, a multimodal agent benchmark focusing on language communication between multiple agents (Section 3).

- Using COMMA, we carefully record conversations and performance metrics between state-of-the-art multimodal models such as QWenVL, InternVL, GPT-4o, GPT-4o1, etc (Section 4).

- We categorize the agent capabilities tested in our model and common failure modes, providing insight into future research directions for improving inter-agent communication (Section 5).

## 2 RELATED WORK

**Multi-agent Frameworks:** There are many emergent agent collaboration works (Gurcan, 2024; Park et al., 2023; Hong et al., 2024; Liu et al., 2024; Ghafarollahi & Buehler, 2024; Li et al., 2023; Wu et al., 2023) among multiple language agents. Multi-agent systems arise mainly in two different scenarios: (1) *role-playing different task executors* (e.g., software development requiring different roles of agents, such as program manager, software architect, programmer Du et al. (2024); Qian et al. (2024); Hong et al. (2024), scientific discovery simulation Wu et al. (2023), and social simulation Park et al. (2023); Gurcan (2024)); (2) *communicating between agents with different pieces of information* Wu et al. (2023); Li et al. (2023) (e.g., consulting experts without sharing some sensitive or confidential data. In our case, the AI solver has some private multimodal data, and the AI expert has domain-specific knowledge or instructions).

**Instruction-based Agent Benchmarks:** Instruction-based agent benchmarks evaluate an agent's capability of following a human instructions to finish a task (e.g., navigating on a website, interacting with an operating system Xu et al. (2024); Xie et al. (2024); Cao et al. (2024)). However, our benchmark focuses more on a communication-based evaluation where two clients engage in multi-turn conversations to solve a task collaboratively.

## 3 BENCHMARK

### 3.1 DESIGN PRINCIPLES OF THE BENCHMARK

Our benchmark is inspired by the cooperative gameplay scenario in Keep Talking and Nobody Explodes Games (2015). In this game, two players work together to defuse a bomb under time pressure. One player, the defuser, can see the bomb but lacks the instructions to disarm it. The other player, the expert, has access to the bomb's manual but cannot see the bomb itself. The players must rely on effective communication to exchange information, navigate challenges, and defuse the bomb.

We generalize this dynamic for our benchmark, shifting the focus to solving vision-language puzzles in a communication-based agent framework. As multimodal agent systems gain momentum, our goal is to create a benchmark that rigorously evaluates their reasoning, communication, and collaborative abilities. The design of our benchmark revolves around the following core principles:

**Cognitive Science:** Our benchmark draws inspiration from foundational principles of intelligence, often defined as the ability to learn from experience, adapt to the environment, and solve problems using cognitive skills Kempf-Leonard (2005). Cognitive Science research has demonstrated that even simple tests can effectively measure cognitive ability Davidson et al. (2006); St Clair-Thompson & Gathercole (2006). Standardized intelligence tests, such as MENSA MENSA International (n.d.) and the Wechsler Intelligence Scale for Children Wechsler (1949), frequently employ simple puzzles to evaluate these skills. Building on this approach, our benchmark aims to assess the core cognitive capabilities of multimodal agents by creating simple vision-language puzzles tailored to test these abilities.

**Language communication:** A critical aspect of our benchmark is evaluating natural language communication between agents. Similar to how players in the original game exchange information verbally, agents in our framework must use language to share observations, clarify ambiguities, and reason about tasks. In order for the agents to succeed, they must display clarity, efficiency, and depth of communication, making it an essential factor in task completion.

**Multi-agent collaboration:** In our benchmark, agents must work together, much like the two-player dynamic of Keep Talking and Nobody Explodes. The collaboration element ensures that tasks require mutual dependency, where each agent contributes unique information or capabilities that are critical to solving the puzzle. This principle highlights how well agents can cooperate and leverage each other's strengths.

**Multimodality:** Our benchmark emphasizes the integration of multiple sensory inputs and outputs, such as vision, language, and audio. The puzzles involve visual elements that agents must

perceive, describe, and interpret, alongside linguistic interactions. This principle assesses an agent's ability to handle and synthesize multimodal information, a skill crucial to real-world applications.

## 3.2 CATEGORIES OF AGENT CAPABILITY

We benchmark agents working under different roles to solve various tasks in multiple settings, each requiring different capabilities. Specifically, the Solver agent must demonstrate strong instruction-following and multimodal reasoning, while the Expert agent is expected to excel in long text summarization and information retrieval. Both agents must possess visual comprehension and descriptive skills to succeed. Below, we outline the core capabilities tested in our benchmark.

**Memory Recall (MR)** In many puzzles, agents must remember their previous actions to progress. This ability is also implicitly tested when agents make mistakes. A competent agent should recall instances where past actions led to errors and adapt to avoid repeating them. The capacity to learn from mistakes and leverage memory is crucial for effective problem-solving in real-world situations.

**Multimodal Grounding (MG)** Since the solver agent can only communicate with the expert with text, it must be able to ground relevant spans of the expert's instructions to the image it currently sees. This grounding of language in visual context is essential for interpreting and following guidance from the expert agent effectively.

**Multi-Step Reasoning (MSR)** Certain puzzles require agents to follow a sequence of actions based on step-by-step reasoning. Much like real-world tasks, such as following a recipe or placing an online order, each action must be deliberate and contribute toward the overall goal. Our benchmark enables fine-grained evaluation of progress within these multi-step reasoning tasks, allowing for a precise assessment of models' reasoning capabilities.

**Real-time Reaction (RT)** Some puzzles challenge agents to process information rapidly and act with precise timing. This is a critical skill for embodied agents operating in dynamic, real-world environments where precise timing and quick reactions are vital.

## 3.3 TASKS

We create 10 puzzles across 4 different categories briefly summarized below. A more comprehensive description along with example images and instruction manuals can be found in Appendix A.

- **ButtonPuzzle (RT ):** The solver must hold a colored button for a specific number of seconds based on the button's color, the strip's color when pressed, and the time remaining on a timer.
- **ColorPuzzle (MR , MSR ):** The solver presses squares of the least common color in a 4x4 grid, then follows a sequence based on a table, aiming to turn all squares white.
- **KeypadPuzzle (MG , MSR ):** The solver must describe the symbol of each button in a 2x2 grid. The expert must then identify a column in the manual containing these four unique symbols and tell the solver to press the symbols in the correct order from top to bottom.
- **LedPuzzle (MR , MSR ):** The solver presses a button if the value of its letter, when multiplied by a stage's LED color multiplier and taken modulo 26, matches the value of the letter diagonally opposite it. At each stage, the letters on the buttons change.
- **MazePuzzle (MG , MSR ):** The solver navigates a mouse through a maze to a colored sphere, pressing the correct button to disarm the module based on the maze's layout.
- **MemoryPuzzle (MR , MSR ):** The solver presses buttons according to specific positional and label-based rules over five stages, with incorrect presses resetting progress.
- **PasswordPuzzle (MG , MSR ):** The solver cycles through letters to form a valid word from a predefined list, submitting the correct word to complete the puzzle.
- **DogPuzzle (RT ):** The solver is presented with an image containing 0-4 dogs. Based on the number of dogs in the image, the solver must press the submit button when the last digit of the timer matches the number of dogs in the image.
- **WhoPuzzle (MG ):** The solver must tell the value on a display to the expert, who will identify a button label. The solver must then tell this label to the expert, and then press the correct button based on a detailed list of instructions.

- **WirePuzzle (MG ):** The solver must cut one of the wires on the display. There are 3 to 6 colored wires, and the correct wire to cut changes depending on the number and order of colors.

# 4 EVALUATION

## 4.1 EXPERIMENTAL SETUP

In this section, we describe the experimental settings of our multi-agent interaction environment where two distinct agents, namely the Solver agent and the Expert agent, engage in iterative dialogue sessions. The primary aim of this setup is to assess the collaborative problem-solving capabilities between different agents or multimodal large language models (MLLMs). During our experiments, we limit the number of conversation turns to 20, allowing for a unified and systematic assessment of interactions. We use greedy decoding when available to maintain consistent agent output across runs and run inference on a single NVIDIA A100 GPU with 80GB RAM. We parse the solver's chosen actions at each conversation turn using exact string matching, and use PyAutoGUI (Sweigart, 2023) to directly perform the action on the interface if the solver outputs a valid action. Our exact prompts for both the solver and expert agent can be found in Appendix D.

## 4.2 EVALUATION METRICS

We meticulously recorded several key performance metrics through multiple iterations of the experiments described below:

- **Success Rate (SR):** The solver agent is assigned a 0 or 1 value for each puzzle depending on if it completed it. These values are averaged across all puzzles to obtain the success rate.
- **Partial Success Rate (PSR):** Because our benchmark includes puzzles with multi-step reasoning, some puzzles can have a more precise success rate evaluation. For these multi-step puzzles, we assign the solver a number between 0 and 100 to indicate its progress towards the solution, and average this number across puzzles to obtain partial success rate. For single-step puzzles, partial success rate is identical to success rate.
- **Average Mistakes (AM):** After an action is chosen by the solver, the environment checks if the action was a mistake. We tally up the mistakes made during each puzzle and take a global average across puzzles to obtain average mistakes.
- **Average Conversation Length (ACL):** We count the number of conversation turns the Solver took to arrive at the solution, or default to the maximum of 20 if the solver failed. This count is averaged across all puzzles to get Average Conversation Length.

## 4.3 MODELS

**Open-Source Models**

- **Human:** We conduct several experiments in which a human plays as the solver or expert to provide a strong baseline. As hiring participants was prohibitively expensive and time consuming, we role played as agents ourselves across 30 sampled puzzles as a preliminary study, and leave further human participation to future work.
- **InternVL** (Chen et al., 2024): A vision-language model by Shanghai AI Lab, designed for cross-modal tasks like visual question answering and image-text retrieval. We evaluate both the 26b and 8b variants of the model.
- **QwenVL** (Bai et al., 2023): We use version 2 of QWenVL (QWen-VL2), offering enhanced pretraining for improved performance on vision-language tasks. We use the 2b and 7b variants.

**Closed-Source Models**

- **GPT-4V**: A version of OpenAI's GPT-4, GPT-4V incorporates visual processing, enabling it to interpret both text and images.
- **GPT-4o**: An optimized, faster, and more cost-effective variant of GPT-4, used for applications requiring speed and efficiency.

- **GPT-4o1**: The most recent version of OpenAI's GPT series models, which claims to have improved reasoning capability via internal chain of thought.

# 5 RESULTS AND ANALYSIS

| Solver | Expert | Average Partial Success Rate % (↑) | | | | | | | | | | |
|---|---|---|---|---|---|---|---|---|---|---|---|---|
| | | Button | Dog | Wire | Who | LED | Memory | Keypad | Password | Color | Maze | Overall |
| GPT4V | Human | 100 ± 0 | 100 ± 0 | 100 ± 0 | 100 ± 0 | 60 ± 39 | 80 ± 42 | 100 ± 0 | 33 ± 47 | 19 ± 13 | 31 ± 20 | 74 ± 38 |
| GPT4o | | 67 ± 50 | 100 ± 0 | 100 ± 0 | 100 ± 0 | 93 ± 41 | 73 ± 46 | 100 ± 0 | 100 ± 0 | 35 ± 18 | 0 ± 0 | 77 ± 32 |
| InternVL8b | | 100 ± 0 | 100 ± 0 | 100 ± 0 | 100 ± 0 | 67 ± 47 | 47 ± 41 | 100 ± 0 | 67 ± 47 | 4 ± 3 | 0 ± 0 | 69 ± 47 |
| GPT4o1 | Human | 100 | 100 | 100 | 100 | 100 | 100 | 0 | 0 | 44 | 100 | 74 |
| GPT4V | | 67 ± 47 | 100 ± 49 | 67 ± 47 | 100 ± 0 | 0 ± 0 | 100 ± 0 | 67 ± 47 | 100 ± 0 | 19 ± 9 | 67 ± 47 | 71 ± 47 |
| GPT4o | | 100 ± 0 | 100 ± 0 | 100 ± 0 | 100 ± 0 | 33 ± 27 | 67 ± 47 | 0 ± 0 | 0 ± 0 | 19 ± 9 | 100 ± 0 | 62 ± 49 |
| InternVL8b | | 100 ± 0 | 100 ± 0 | 50 ± 0 | 100 ± 0 | 50 ± 47 | 10 ± 41 | 62 ± 0 | 50 ± 47 | 0 ± 5 | 50 ± 0 | 55 ± 47 |

Table 1: Average Partial Success Rate of multimodal agents on each puzzle with Human as one agent. The solver is assigned a value between 0-100 indicating how far the solver progressed through the puzzle. The partial success rate is calculated by averaging this value over 10, 3, and 100 independent runs of each puzzle for the AI-AI, AI-Human, and random settings. The overall column is an average across all the puzzles.

| Solver | Expert | Average Partial Success Rate % (↑) | | | | | | | | | | |
|---|---|---|---|---|---|---|---|---|---|---|---|---|
| | | Button | Dog | Wire | Who | LED | Memory | Keypad | Password | Color | Maze | Overall |
| Random | InternVL | 100 ± 0 | 100 ± 0 | 100 ± 0 | 100 ± 0 | 85 ± 31 | 25 ± 14 | 33 ± 20 | 0 ± 0 | 1 ± 1 | 15 ± 9 | 56 ± 41 |
| GPT4V | GPT4V | 80 ± 40 | 60 ± 49 | 100 ± 0 | 90 ± 30 | 68 ± 42 | 24 ± 26 | 72 ± 43 | 0 ± 0 | 14 ± 13 | 21 ± 31 | 53 ± 46 |
| GPT4o | GPT4o | 100 ± 0 | 100 ± 0 | 100 ± 0 | 90 ± 30 | 26 ± 18 | 34 ± 28 | 40 ± 43 | 0 ± 0 | 14 ± 10 | 0 ± 0 | 50 ± 45 |
| InternVL | GPT4o | 100 ± 0 | 100 ± 0 | 100 ± 0 | 100 ± 0 | 50 ± 47 | 30 ± 24 | 56 ± 38 | 7 ± 9 | 16 ± 11 | 0 ± 0 | 46 ± 29 |
| InternVL26b | InternVL26b | 90 ± 30 | 100 ± 0 | 80 ± 40 | 20 ± 40 | 30 ± 46 | 20 ± 30 | 10 ± 0 | 0 ± 0 | 0 ± 0 | 0 ± 0 | 35 ± 48 |
| InternVL8b | InternVL8b | 100 ± 0 | 100 ± 0 | 30 ± 38 | 40 ± 43 | 4 ± 9 | 12 ± 23 | 5 ± 8 | 0 ± 0 | 11 ± 7 | 0 ± 0 | 30 ± 31 |
| InternVL8b | QwenVL7b | 100 ± 0 | 100 ± 0 | 20 ± 40 | 20 ± 40 | 56 ± 43 | 20 ± 15 | 15 ± 23 | 0 ± 0 | 7 ± 8 | 25 ± 21 | 36 ± 40 |
| QwenVL2b | QwenVL2b | 100 ± 0 | 100 ± 0 | 30 ± 46 | 20 ± 40 | 100 ± 0 | 0 ± 0 | 17 ± 25 | 0 ± 0 | 9 ± 6 | 2 ± 2 | 38 ± 43 |
| QwenVL7b | GPT4o | 90 ± 30 | 90 ± 30 | 55 ± 50 | 30 ± 46 | 16 ± 17 | 26 ± 31 | 35 ± 38 | 0 ± 0 | 7 ± 8 | 9 ± 13 | 36 ± 41 |
| QwenVL7b | InternVL8b | 100 ± 0 | 100 ± 0 | 40 ± 49 | 30 ± 46 | 0 ± 0 | 34 ± 31 | 5 ± 8 | 0 ± 0 | 6 ± 8 | 13 ± 13 | 33 ± 39 |
| QwenVL7b | QwenVL7b | 90 ± 30 | 90 ± 30 | 30 ± 46 | 10 ± 30 | 52 ± 37 | 0 ± 0 | 33 ± 23 | 0 ± 0 | 4 ± 5 | 12 ± 16 | 32 ± 38 |

Table 2: Average Partial Success Rate of multimodal agents on each puzzle without Human as one agent. The solver is assigned a value between 0-100 indicating how far the solver progressed through the puzzle. The partial success rate is calculated by averaging this value over 10, 3, and 100 independent runs of each puzzle for the AI-AI, AI-Human, and random settings. The overall column is an average across all the puzzles.

## 5.1 OVERALL PERFORMANCE

Table 2 illustrates the performances of all combinations of solver and expert pairs we evaluated. We evaluate some combinations of different open-source models because they are free, and leave pairings of separate closed-source models for future work. Intrestingly, increasing model size does not always improve performance, as QWenVL 2b (38% PSR) outperforms QWenVL 7b (32% PSR). We observe that the random agent performs well on simple puzzles which only require one correct action (e.g. 100% on WirePuzzle, 100% on WhoPuzzle) and even outperforms all of the AI-AI setting performances. However, a random agent struggles with puzzles which require several correct actions in a row, as evidenced by its 0 percent success rate and low partial success rate in Memory, Maze, Password, Keypad, and Color puzzles.

Our results show that GPT-4o1 is the most powerful agent to use, although we only run a single iteration of each puzzle with a Human-AI setting due to rate limit issues. It makes the most progress in the color puzzle, achieving a partial success rate of 44%, followed by GPT-4o at 35%. In this puzzle, we observed that GPT-4o1's "reasoning process" made correct connections with different parts of the puzzle's manual, though sometimes it made incorrect conclusions at the end of the process. GPT-4V is the second best agent, with an average partial score of 53% in the AI-AI setting, although the performance is still far worse than human performance. For example, switching from a gpt-4o solver to a human solver increases overall partial success rate from 53% to 74%.

## 5.2 QUALITATIVE ANALYSIS ON MODEL FAILURES

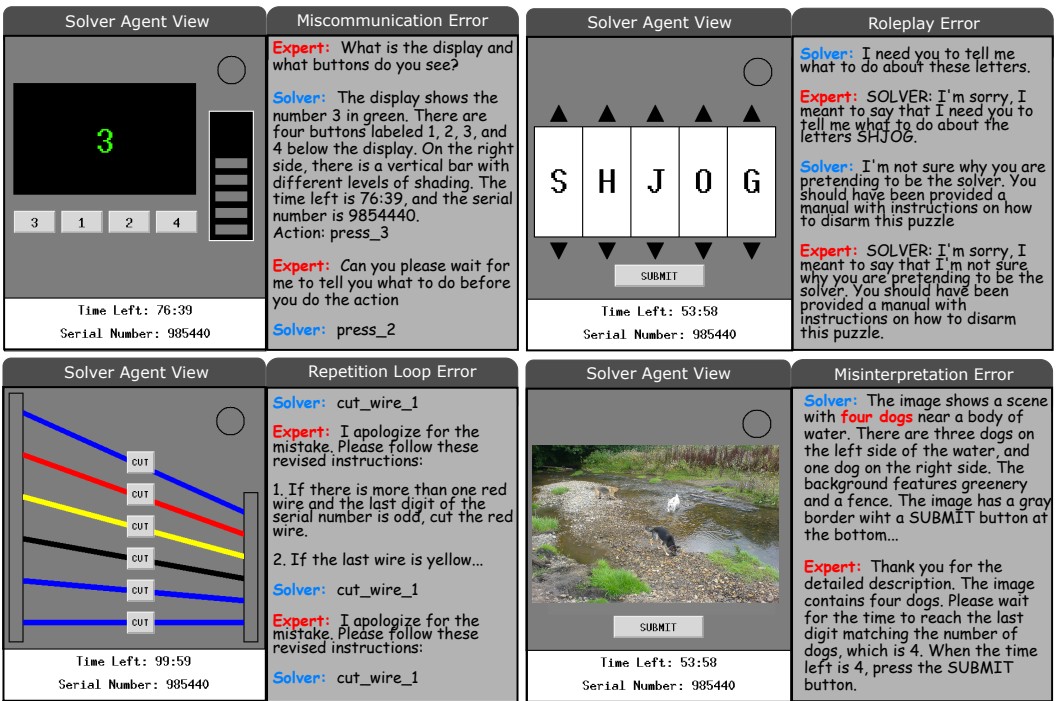

Figure 2: Case study examples of with the InternVL 8b model playing the role of AI solver and AI expert, resulting in failure to complete the task. **Top Left**: An AI solver **Miscommunication** error causes it to ignore instructions from the human expert, causing it to try out actions without understanding the solution. **Top Right**: The AI expert misunderstands its role with a **Roleplay** error and pretends it is the Solver. **Bottom Left**: The solver repeats the same bad action, resulting in a **Repetition Loop** error. **Bottom Right**: The solver misinterprets the number of dogs in the image, leading to a **Misinterpretation** error.

In this section, we highlight key takeaways and common failure modes displayed by the agents during their conversations. We manually classify errors across 50 conversations into the following

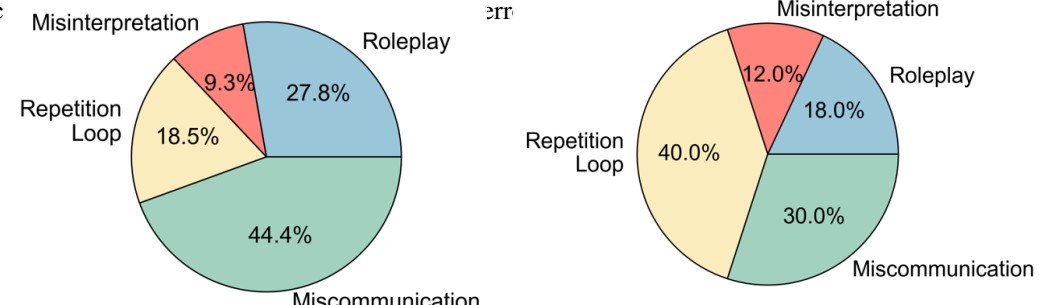

Figure 3: Distribution of error category across all puzzles in AI-AI (GPT-4o) setting.

Figure 4: Distribution of error category across all puzzles in the AI-AI (InternVL26B) setting.

- **Roleplay:** The expert thinks it is the solver or vice versa. Figure 2 illustrates how the expert can misunderstand its role assignment, leading to miscommunication and failure to solve the puzzle.

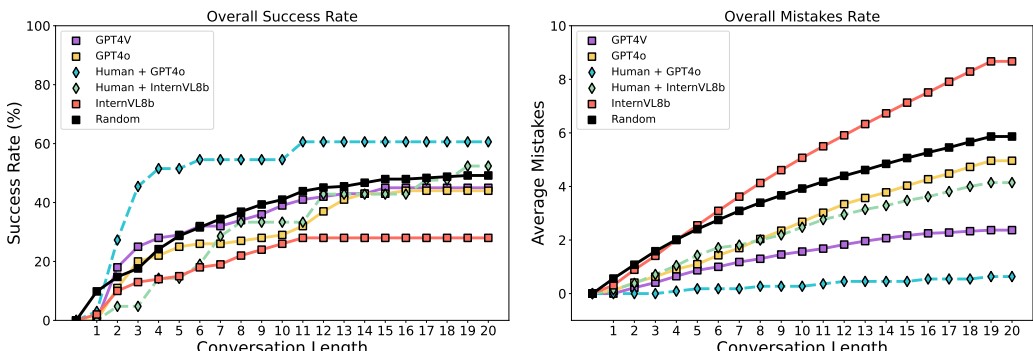

Figure 5: We plot the overall success and mistake rate on our benchmark as a function of the number of allowed conversation turns. We obtain the overall success rate and mistake rate by averaging over 100 sampled instances across all puzzles for the AI-AI setting, and 30 sampled instances for the AI-Human setting. Diamond marker indicates a human is the solver agent. Square marker indicates the solver and expert are played by the same AI model. Random is a baseline where the solver agent chooses actions uniformly at random at each time step.

- **Misinterpretation:** The solver misunderstands the current puzzle state/signal, resulting in failure. For instance, Figure 2 showcases the solver misinterpreting the number of dogs in the image, leading to incorrect instructions from the expert.
- **Repetition Loop:** The solver sometimes repeats its past incorrect actions, even if is in a situation it has encountered before. We classify any repeated incorrect state, action pair into this category.
- **Miscommunication:** As shown in Figure 2, the agent occasionally does not listen to the expert's instructions, attempting to solve the puzzle on its own as if it were the expert. We also observed some open source models such as LLaVA don't have instruction following capability for this task without further finetuning. Additionally, the solver sometimes describes the puzzle incorrectly to the expert which results in failure.

**Repeated Actions are a Common Failure for Agents** Both open-source and closed-source models often fail due to repeating bad actions. As shown in Figure 4, InternVL is more inclined to get stuck in a repetition loop compared to GPT-4o (40% vs 18.5%). This suggests a potential improvement direction by including multi-step repetitive conversations during training, in which the model should learn to break out of the loop.

**Agents Make More Miscommunication Errors than Misinterpretation** We observe that misinterpretation accounts for a much smaller proportion of total errors compared to miscommunication related errors (9.3% vs 44.5%) for GPT-4o and (12.0% vs 30.0%) for InternVL. We hypothesize this occurs because the training data mixture for these models primarily includes high quality single-agent data from academic benchmarks such as Visual Question Answering (Antol et al., 2015), Image Captioning, etc. Including tasks requiring communication may help address this issue.

## 5.3 Fine-grained Analysis

**Multimodal Agents Struggle to Learn from Past Mistakes** An important skill for humans is to learn from past mistakes to adapt to new situations. Here we analyze if agents can display a similar capability and recover when exploring a bad trajectory when solving a puzzle. Figure 5 plots the number of allowed conversation turns to solve a puzzle, along with the overall success and mistakes rate of several multimodal agents. We note the following observations. First, incorporating a human in the pipline in the form of a human solver significantly improves overall success rate, being the only agent to achieve over random baseline performance at the 20-turn conversation mark. This is also supported by the mistakes plot, in which the human solver setting generally displays lower mistakes as the conversation progresses compared to the full AI setting. In fact, the human solver, gpt-4o expert setting shows zero mistakes over the course of most conversations, with the main reason for failure being the conversation limit. Second, humans appear to have greater ability to

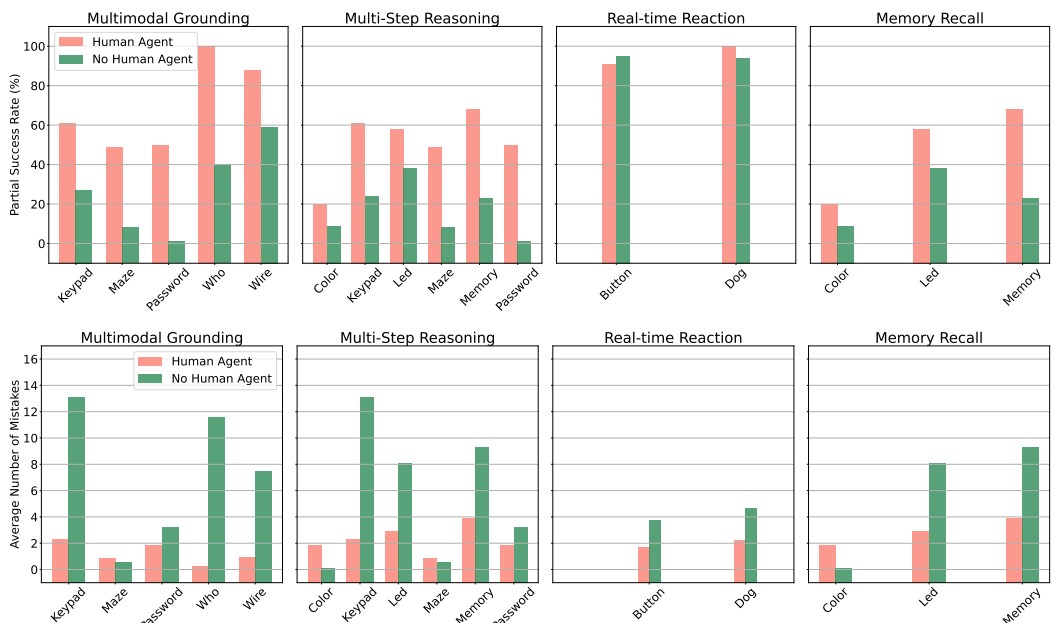

Figure 6: Average partial success rate (**Top**) and mistake rate (**Bottom**) for puzzles based on categories with and without human involvement. Each bar is an average performance across all model combinations. Note, some puzzles fall into multiple categorie and appear in multiple subplots.

recover, as indicated by the faster increase in their success rate as conversation length increases, as well as the fact that they make less mistakes over time in the mistakes chart.

**Performance Based on Capability**   Here we group the model performance based on the category tested: Memory (MR), Grounding (MG), Reasoning (MSR), and Reaction (RT).

**Multimodal Agents Excel at Simple Realtime Tasks**   Figure 6 gives a more nuanced look at how well multimodal agents are equipped to deal with puzzles of different nature. The agents performed well in the RT category, with the Button puzzle having the highest average partial score at 95% PSR, and Dog puzzle follows closely at 94% PSR. This suggests that agents are best at real-time reaction tasks which may involve quick decision-making based on immediate visual input and not much further communication.

**Grounding is a Challenge for Multimodal Agents**   Multimodal grounding presents a significant challenge to agents in the AI-AI setting, as seen in the varied performance. This ability requires agents to interpret and connect visual stimuli with textual instructions. The stronger performance on puzzles like Wire puzzle (88% PSR) and Who puzzle (100% PSR) indicates that agents manage better when tasks are more structured or involve simpler visual-text connections. In contrast, puzzles like the Password puzzle (1% PSR) and Maze puzzle (8% PSR), which are more abstract or less structured, present greater difficulties.

**Multimodal Agents Struggle with Memory and Multi-Step Reasoning**   Memory-based puzzles present a challenge for agents. While the LED Puzzle (38% PSR) shows moderate performance, the Color puzzle highlights a significant difficulty (9% PSR). This suggests that agents may struggle with tasks requiring them to remember previous states and actions to progress or solve sequential problems efficiently like the Memory puzzle (23% PSR). The complexity of multi-stage memory tasks could explain the poor performance. In the same vein, multi-step logical reasoning puzzles require agents to think ahead and execute a series of steps to achieve the final goal. The low performance on the Color puzzle (9% PSR) and KeyPad puzzle (24% PSR) suggests that complex reasoning tasks, especially those involving multiple stages, remain a significant challenge for agents.

## 6 CONCLUSION

In this paper, we address a critical gap in the field of multi-modal agents by introducing a novel benchmark specifically designed to evaluate communication in a multi-modal, multi-agent system. While substantial progress has been made in developing individual multi-modal agents, collaborative frameworks remain under-explored, particularly in scenarios requiring secure communication and the handling of sensitive data. Our benchmark aims to bridge this gap by simulating real-world conditions where agents possess complementary information and must work together to achieve complex goals. We comprehensively evaluate metrics such as partial success rate, mistake rate, and document common failure modes for both AI-AI and AI-Human interactions. Our findings show that multimodal agents struggle to communicate with each other, often falling short of even a simple random baseline due to poor communication and frequently repeated bad actions. These findings emphasize the need for deeper investigation into enhancing inter-agent collaboration. We hope the insights from our benchmark lay the foundation for future research on multi-modal agent collaboration and inspires the community to explore innovative approaches to improve multimodal agent capabilities this emerging field of communicative multimodal systems.

## 7 LIMITATIONS

While we aim to construct a holistic framework for multimodal agent communication, our experiments may not represent all possible scenarios in our puzzles. We conduct a preliminary study by sampling puzzle configurations and conversations between agents, and we leave more comprehensive evaluation and expansion of puzzle categories to future work. Additionally, there will inevitably be a simulation-to-reality gap from our benchmark to real-world situations, thus a high score on our benchmark may not perfectly generalize to real-world communication scenarios. Lastly, we acknowledge that there is inherent risk to using multimodal agents when handling private data. Given that LLMs have been shown to be prone to jailbreaking, it is critical to take additional safety measures before deploying an agent in practice, even if it achieves a high score on our benchmark.

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

## A  MANUALS

## BUTTON PUZZLE

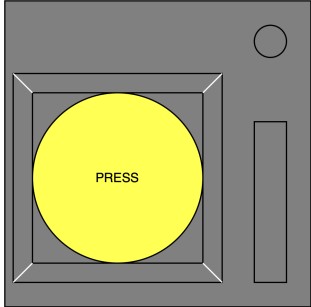

If the button is yellow, hold the button and refer to the next set of instructions of when to release it.

If you start holding the button down, a colored strip will light up on the right side of the module. Based on its color, you must release the button at a specific point in time:

- **Blue strip**: release when the countdown timer has a **4** in any position.
- **White strip**: release when the countdown timer has a **1** in any position.
- **Yellow strip**: release when the countdown timer has a **5** in any position.
- **Any other color strip**: release when the countdown timer has a **1** in any position.

COLOR PUZZLE

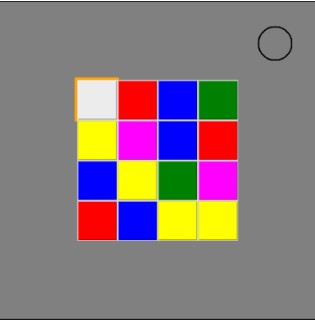

Press all squares in the correct group to progress the module. Pressing a square will cause it to light up white. Make all squares white to disarm the module.

To begin, press the color group containing the fewest squares. If there is a tie, you should choose the first color that appears in the list:

- Red
- Blue
- Green
- Yellow
- Magenta

Then use the table to determine the next group to press in each stage. "Group" refers to all squares of a particular color, or all non-white squares in the topmost row or leftmost column containing non-white squares. Pressing an incorrect square will result in a strike and reset the module. White squares will remain white for the duration of the module, but non-white squares may change color in each stage.

The table below helps to choose the next subgroup to press. The numbered keys correspond to the number of currently white squares, and the "previously pressed color" key gives you values that indicate what color to press next based on the corresponding number of white squares.

**Previously Pressed Color**: {Red, Blue, Green, Yellow, Magenta, Row, Column}

$$1 : \{Blue, Column, Red, Yellow, Row, Green, Magenta\}$$
$$2 : \{Row, Green, Blue, Magenta, Red, Column, Yellow\}$$
$$3 : \{Yellow, Magenta, Green, Row, Blue, Red, Column\}$$
$$4 : \{Blue, Green, Yellow, Column, Red, Row, Magenta\}$$
$$5 : \{Yellow, Row, Blue, Magenta, Column, Red, Green\}$$
$$6 : \{Magenta, Red, Yellow, Green, Column, Blue, Row\}$$
$$7 : \{Green, Row, Column, Blue, Magenta, Yellow, Red\}$$
$$8 : \{Magenta, Red, Green, Blue, Yellow, Column, Row\}$$
$$9 : \{Column, Yellow, Red, Green, Row, Magenta, Blue\}$$
$$10 : \{Green, Column, Row, Red, Magenta, Blue, Yellow\}$$
$$11 : \{Red, Yellow, Row, Column, Green, Magenta, Blue\}$$
$$12 : \{Column, Row, Column, Row, Row, Column, Row\}$$
$$13 : \{Row, Column, Row, Column, Row, Column, Column\}$$
$$14 : \{Column, Column, Row, Row, Column, Row, Column\}$$
$$15 : \{Row, Row, Column, Row, Column, Column, Row\}$$

KEYPAD PUZZLE

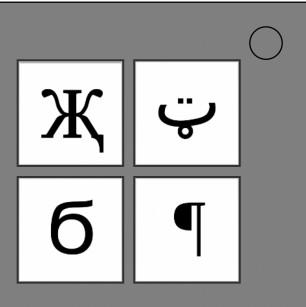

Only one column has all four symbols from the keypad. Press the four buttons in the order their symbols appear from top to bottom within that column.

LED PUZZLE

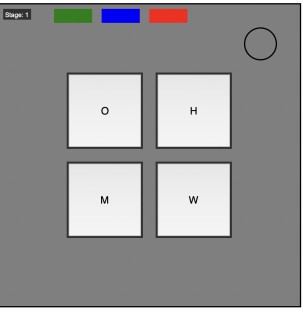

Two to five LEDs are installed at the top of the module, representing stages. To disarm the module, these stages must be solved in order. Four buttons with four different letters are shown. The letters change at each stage. The current stage is indicated by a number in the top left of the module. The current stage's multiplier is indicated by that stage's LED according to the following mapping:

- Red: 2

- Green: 3

- Blue: 4

- Yellow: 5

- Purple: 6

- Orange: 7

Assign each letter of the alphabet to the numbers 0-25 (A = 0, B = 1, C = 2, etc.). A button is correct if its letter value, multiplied by the current stage's multiplier, modulo 26, is equal to the regular value of the letter on its diagonally opposite button. At each stage, press a correct button. There may be more than one possible answer.

MAZE PUZZLE

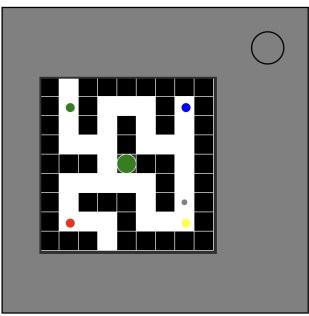

The mouse is the grey sphere. It can only move into other white squares. Dark squares are walls and it cannot move into those. The mouse can move forward or backward or turn left or right. To disarm the module, navigate the mouse to the accepting position and press the circular button with the labyrinth. Pressing the button at any other location causes a strike. The accepting position is marked with one of four colored spheres. Which one depends on the color of the torus in the middle of the maze, according to the table below.

- **Torus Colors**: Green, Blue, Red, Yellow
- **Sphere Colors**: Blue, Red, Green, Yellow

MEMORY PUZZLE

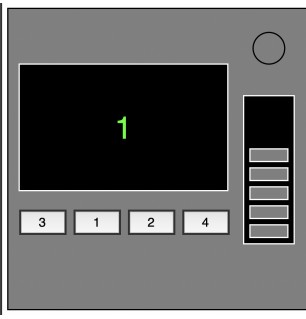

Press the correct button to progress the module to the next stage. Complete all stages to disarm the module. Pressing an incorrect button will reset the module back to stage 1. Button positions are ordered from left to right.

STAGE 1

- If the display is 1, press the button in the second position.
- If the display is 2, press the button in the second position.
- If the display is 3, press the button in the third position.
- If the display is 4, press the button in the fourth position.

STAGE 2

- If the display is 1, press the button labeled "4".
- If the display is 2, press the button in the same position as you pressed in stage 1.
- If the display is 3, press the button in the first position.
- If the display is 4, press the button in the same position as you pressed in stage 1.

STAGE 3

- If the display is 1, press the button with the same label you pressed in stage 2.

- If the display is 2, press the button with the same label you pressed in stage 1.

- If the display is 3, press the button in the third position.

- If the display is 4, press the button labeled "4".

STAGE 4

- If the display is 1, press the button in the same position as you pressed in stage 1.

- If the display is 2, press the button in the first position.

- If the display is 3, press the button in the same position as you pressed in stage 2.

- If the display is 4, press the button in the same position as you pressed in stage 2.

STAGE 5

- If the display is 1, press the button with the same label you pressed in stage 1.

- If the display is 2, press the button with the same label you pressed in stage 2.

- If the display is 3, press the button with the same label you pressed in stage 4.

- If the display is 4, press the button with the same label you pressed in stage 3.

## PASSWORD PUZZLE

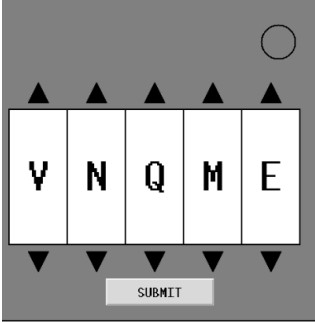

The buttons above and below each letter will cycle through the possibilities for that position. Each cycle will have 3 consecutive letters. Only one combination of the available letters will match a password from the list below. Press the submit button once the correct word has been set.

LIST OF POSSIBLE WORDS:

- about, after, again, below, could, every, first, found, great, house, large, learn, never, other, place, plant, point, right, small, sound, spell, still, study, their, there, these, thing, think, three, water, where, which, world, would, write.

WHO PUZZLE

1. Read the display and use step 1 to determine which button label to read. 2. Using this button label, use step 2 to determine which button to push.

STEP 1:

Based on the display, ask the SOLVER to read the label of a particular button and proceed to step 2:

- "YES": Middle Left
- "FIRST": Top Right
- "DISPLAY": Bottom Right
- "OKAY": Top Right
- "SAYS": Bottom Right
- "NOTHING": Middle Left
- "(No Text)": Bottom Left
- "BLANK": Middle Right
- "NO": Bottom Right
- "LED": Middle Left
- "LEAD": Bottom Right
- "READ": Middle Right
- "RED": Middle Right
- "REED": Bottom Left
- "LEED": Bottom Left
- "HOLD ON": Bottom Right
- "YOU": Middle Right
- "YOU ARE": Bottom Right
- "YOUR": Middle Right
- "YOU'RE": Middle Right
- "UR": Top Left
- "THERE": Bottom Right
- "THEY'RE": Bottom Left
- "THEIR": Middle Right
- "THEY ARE": Middle Left
- "SEE": Bottom Right
- "C": Top Right
- "CEE": Bottom Right

STEP 2:

Using the label from step 1, push the first button that appears in its corresponding list:

- "READY": YES, OKAY, WHAT, MIDDLE, LEFT, PRESS, RIGHT, BLANK, READY, NO, FIRST, UHHH, NOTHING, WAIT

- "FIRST": LEFT, OKAY, YES, MIDDLE, NO, RIGHT, NOTHING, UHHH, WAIT, READY, BLANK, WHAT, PRESS, FIRST

- "NO": BLANK, UHHH, WAIT, FIRST, WHAT, READY, RIGHT, YES, NOTHING, LEFT, PRESS, OKAY, NO, MIDDLE

- "BLANK": WAIT, RIGHT, OKAY, MIDDLE, BLANK, PRESS, READY, NOTHING, NO, WHAT, LEFT, UHHH, YES, FIRST

- "NOTHING": UHHH, RIGHT, OKAY, MIDDLE, YES, BLANK, NO, PRESS, LEFT, WHAT, WAIT, FIRST, NOTHING, READY

- "YES": OKAY, RIGHT, UHHH, MIDDLE, FIRST, WHAT, PRESS, READY, NOTHING, YES, LEFT, BLANK, NO, WAIT

- "WHAT": UHHH, WHAT, LEFT, NOTHING, READY, BLANK, MIDDLE, NO, OKAY, FIRST, WAIT, YES, PRESS, RIGHT

- "UHHH": READY, NOTHING, LEFT, WHAT, OKAY, YES, RIGHT, NO, PRESS, BLANK, UHHH, MIDDLE, WAIT, FIRST

- "LEFT": RIGHT, LEFT, FIRST, NO, MIDDLE, YES, BLANK, WHAT, UHHH, WAIT, PRESS, READY, OKAY, NOTHING

- "RIGHT": YES, NOTHING, READY, PRESS, NO, WAIT, WHAT, RIGHT, MIDDLE, LEFT, UHHH, BLANK, OKAY, FIRST

- "MIDDLE": BLANK, READY, OKAY, WHAT, NOTHING, PRESS, NO, WAIT, LEFT, MIDDLE, RIGHT, FIRST, UHHH, YES

- "OKAY": MIDDLE, NO, FIRST, YES, UHHH, NOTHING, WAIT, OKAY, LEFT, READY, BLANK, PRESS, WHAT, RIGHT

- "WAIT": UHHH, NO, BLANK, OKAY, YES, LEFT, FIRST, PRESS, WHAT, WAIT, NOTHING, READY, RIGHT, MIDDLE

- "PRESS": RIGHT, MIDDLE, YES, READY, PRESS, OKAY, NOTHING, UHHH, BLANK, LEFT, FIRST, WHAT, NO, WAIT

- "YOU": SURE, YOU ARE, YOUR, YOU'RE, NEXT, UH HUH, UR, HOLD, WHAT?, YOU, UH UH, LIKE, DONE, U

- "YOU ARE": YOUR, NEXT, LIKE, UH HUH, WHAT?, DONE, UH UH, HOLD, YOU, U, YOU'RE, SURE, UR, YOU ARE

- "YOUR": UH UH, YOU ARE, UH HUH, YOUR, NEXT, UR, SURE, U, YOU'RE, YOU, WHAT?, HOLD, LIKE, DONE

- "YOU'RE": YOU, YOU'RE, UR, NEXT, UH UH, YOU ARE, U, YOUR, WHAT?, UH HUH, SURE, DONE, LIKE, HOLD

- "UR": DONE, U, UR, UH HUH, WHAT?, SURE, YOUR, HOLD, YOU'RE, LIKE, NEXT, UH UH, YOU ARE, YOU

- "U": UH HUH, SURE, NEXT, WHAT?, YOU'RE, UR, UH UH, DONE, U, YOU, LIKE, HOLD, YOU ARE, YOUR

- "UH HUH": UH HUH, YOUR, YOU ARE, YOU, DONE, HOLD, UH UH, NEXT, SURE, LIKE, YOU'RE, UR, U, WHAT?

- "UH UH": UR, U, YOU ARE, YOU'RE, NEXT, UH UH, DONE, YOU, UH HUH, LIKE, YOUR, SURE, HOLD, WHAT?

- "WHAT?": YOU, HOLD, YOU'RE, YOUR, U, DONE, UH UH, LIKE, YOU ARE, UH HUH, UR, NEXT, WHAT?, SURE

- "DONE": SURE, UH HUH, NEXT, WHAT?, YOUR, UR, YOU'RE, HOLD, LIKE, YOU, U, YOU ARE, UH UH, DONE

- "NEXT": WHAT?, UH HUH, UH UH, YOUR, HOLD, SURE, NEXT, LIKE, DONE, YOU ARE, UR, YOU'RE, U, YOU

- "HOLD": YOU ARE, U, DONE, UH UH, YOU, UR, SURE, WHAT?, YOU'RE, NEXT, HOLD, UH HUH, YOUR, LIKE

- "SURE": YOU ARE, DONE, LIKE, YOU'RE, YOU, HOLD, UH HUH, UR, SURE, U, WHAT?, NEXT, YOUR, UH UH

- "LIKE": YOU'RE, NEXT, U, UR, HOLD, DONE, UH UH, WHAT?, UH HUH, YOU, LIKE, SURE, YOU ARE, YOUR

## WIRE PUZZLE

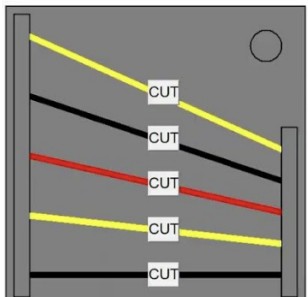

Here is the manual: The WirePuzzle module can have 3-6 wires on it. Only the one correct wire needs to be cut to disarm the module. Wire ordering begins with the first on the top.

### 3 WIRES:

- If there are no red wires, cut the second wire.

- Otherwise, if the last wire is white, cut the last wire.

- Otherwise, if there is more than one blue wire, cut the last blue wire.

- Otherwise, cut the last wire.

### 4 WIRES:

- If there is more than one red wire and the last digit of the serial number is odd, cut the last red wire.

- Otherwise, if the last wire is yellow and there are no red wires, cut the first wire.

- Otherwise, if there is exactly one blue wire, cut the first wire.

- Otherwise, if there is more than one yellow wire, cut the last wire.

- Otherwise, cut the second wire.

### 5 WIRES:

- If the last wire is black and the last digit of the serial number is odd, cut the fourth wire.

- Otherwise, if there is exactly one red wire and there is more than one yellow wire, cut the first wire.

- Otherwise, if there are no black wires, cut the second wire.

- Otherwise, cut the first wire.

6 WIRES:

- If there are no yellow wires and the last digit of the serial number is odd, cut the third wire.
- Otherwise, if there is exactly one yellow wire and there is more than one white wire, cut the fourth wire.
- Otherwise, if there are no red wires, cut the last wire.
- Otherwise, cut the fourth wire.

DOG PUZZLE

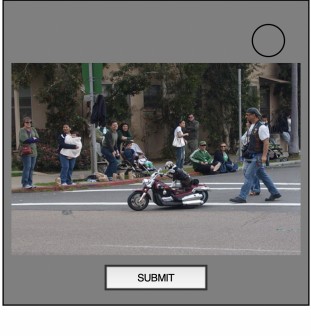

A picture containing 0-5 dogs will be shown on the display. Based on the number of dogs in the image, press the submit button when the last digit of the time left matches the number of dogs in the image.

## B    PUZZLE LISTS

- **ButtonPuzzle:** The solver is presented with an empty strip, a colored button and a timer counting down from 10 minutes. When the button is pressed and held, the strip turns a certain color. Based off of a combination of the color of the button, the color of the strip and the time on the clock, the solver has to keep the button pressed for a certain number of seconds.

- **ColorPuzzle:** The solver is presented with a 4×4 grid of colored tiles. The solver must first identify the color group with the fewest squares on a 4x4 grid and press all the squares of that color to start the module. The solver then needs to refer to a table to determine the next group to press based on the current configuration. Pressing any incorrect square results in a strike and resets the module. Non-white squares may change color after each stage. The goal is to make all squares on the grid white by following the correct sequence of groups.

- **KeypadPuzzle:** The solver has to examine a 2x2 grid of unique symbols and identify which of the four columns below the grid contains all four symbols from the grid. Once the correct column is found, the solver must press the buttons in that column in the order the symbols appear from top to bottom.

- **LedPuzzle:** The solver progresses through 2 to 5 stages, each indicated by an LED color that specifies a multiplier (Red: 2, Green: 3, Blue: 4, Yellow: 5, Purple: 6, Orange: 7). Four buttons with changing letters are shown at each stage. The solver must assign values to letters (A = 0, B = 1, etc.) and press a button if its letter value, when multiplied by the stage's multiplier and taken modulo 26, equals the value of the letter on its diagonally opposite button. Each stage requires pressing a correct button, and there may be multiple valid choices.

- **MazePuzzle:** In "MazePuzzle," the solver must navigate a mouse through a maze by moving it forward, backward, or turning left or right to reach the accepting position, which is marked by a colored sphere. The color of the accepting sphere depends on the color of the

torus in the middle of the maze, with the mapping being Green $\to$ Blue, Blue $\to$ Red, Red $\to$ Green, and Yellow $\to$ Yellow. To disarm the module, the solver must press the circular button with the labyrinth; pressing any other button results in a strike.

- **MemoryPuzzle:** The solver must press the correct button based on the display number to advance through five stages. Incorrect presses reset the module to stage 1. Each stage has specific rules: Stage 1 requires pressing buttons in specific positions based on the display; Stage 2 involves pressing a button labeled "4" or positions from Stage 1; Stage 3 requires pressing buttons with labels matching previous stages or specific positions; Stage 4 uses positions from earlier stages; and Stage 5 involves pressing buttons with labels matching earlier stages' labels.

- **PasswordPuzzle:** The solver cycles through letters above and below each position to form a word. Each cycle displays three consecutive letters, and only one combination will match a predefined list of possible words. Once the correct word is set, the solver must press the submit button to complete the puzzle. The list of possible words includes terms like "about," "after," "great," and "write."

- **SoundPuzzle:** The solver listens to a sound clip and matches it to one of the options provided. Each sound clip is associated with a code made up of four symbols ($, *, &, #). After identifying the correct option from the list (e.g., Taxi Dispatch, Cow, Extractor Fan, Train Station), the solver enters the corresponding code using a four-button keypad to proceed.

- **WhoPuzzle** The solver reads a display to determine which button label to reference and then uses that label to find which button to press based on a predefined list. The process involves two steps: first, the display directs you to a specific button label according to a detailed list of instructions. Second, using that label, you select the appropriate button from a secondary list of options. Successfully following these steps in sequence will advance the module.

- **WirePuzzle:** The solver is presented with between 3 and 6 wires of different colors. Based off of the ordering and number of colors of each type, the solver has to cut the wires in a specific order. The manual lists out the different branches that can be possible for each setting.

## C  ADDITIONAL STATISTICS

| Solver | Expert | Average Number of Mistakes ($\downarrow$) | | | | | | | | | | |
|--------|--------|--------|-----|------|-------|-------|--------|--------|----------|-------|------|---------|
| | | Button | Dog | Wire | Who | LED | Memory | Keypad | Password | Color | Maze | **Overall** |
| | GPT4V | 0.00 | 0.00 | 0.00 | 0.00 | 0.00 | 0.00 | 0.00 | 0.00 | 0.00 | 0.00 | 0.00 |
| Human | GPT4o | 0.00 | 0.00 | 0.00 | 0.00 | 0.00 | 0.00 | 0.00 | 0.00 | 7.00 | 0.00 | 0.64 |
| | InternVL8b | 2.50 | 6.00 | 0.75 | 5.00 | 7.50 | 12.00 | 9.00 | 2.50 | 0.00 | 0.50 | 4.14 |
| GPT4V | | 3.00 | 2.33 | 2.33 | 0.33 | 1.67 | 1.33 | 0.00 | 0.00 | 0.00 | 3.00 | 1.45 |
| GPT4o | Human | 1.67 | 3.67 | 0.00 | 0.33 | 2.00 | 3.67 | 1.33 | 0.00 | 4.00 | 1.67 | 1.83 |
| InternVL8b | | 3.00 | 1.50 | 2.67 | 1.00 | 6.33 | 6.33 | 3.33 | 5.67 | 0.00 | 0.00 | 3.15 |
| GPT4V | GPT4V | 2.30 | 2.40 | 0.10 | 0.60 | 5.50 | 1.30 | 5.40 | 1.40 | 0.00 | 4.70 | 2.37 |
| QwenVL7b | InternVL8b | 5.50 | 4.20 | 5.00 | 13.40 | 0.00 | 6.90 | 10.80 | 0.00 | 0.00 | 0.00 | 4.58 |
| InternVL | GPT4o | 3.70 | 3.80 | 0.00 | 7.60 | 8.40 | 6.70 | 16.00 | 1.60 | 0.00 | 0.00 | 4.78 |
| GPT4o | GPT4o | 4.10 | 5.10 | 0.20 | 2.50 | 6.80 | 3.70 | 12.00 | 15.20 | 0.00 | 0.00 | 4.96 |
| QwenVL7b | GPT4o | 3.00 | 3.30 | 6.55 | 12.10 | 9.60 | 7.00 | 7.90 | 0.00 | 0.00 | 1.40 | 5.10 |
| InternVL26b | InternVL26b | 4.10 | 2.80 | 2.90 | 11.50 | 11.30 | 8.00 | 14.00 | 0.00 | 0.00 | 2.40 | 5.70 |
| Random | InternVL | 4.17 | 3.21 | 2.80 | 3.48 | 9.92 | 14.08 | 14.68 | 2.04 | 0.00 | 3.96 | 5.86 |
| InternVL8b | QwenVL7b | 3.30 | 2.40 | 14.80 | 13.50 | 10.30 | 8.80 | 16.30 | 0.00 | 0.00 | 0.00 | 6.94 |
| QwenVL7b | QwenVL7b | 3.10 | 4.30 | 11.80 | 17.20 | 11.60 | 9.10 | 12.60 | 0.00 | 0.00 | 0.00 | 6.97 |
| QwenVL2b | QwenVL2b | 2.50 | 10.60 | 13.30 | 15.20 | 1.90 | 16.00 | 16.20 | 0.00 | 0.00 | 1.00 | 7.67 |
| InternVL8b | InternVL8b | 4.50 | 3.80 | 12.60 | 11.00 | 13.10 | 17.20 | 12.00 | 12.50 | 0.00 | 0.00 | 8.67 |

Table 3: Average number of mistakes the solver made for various puzzles. We average the mistakes over 10, 3, and 100 independent runs of each puzzle for the AI-AI, AI-Human, and random settings. The overall column is an average across all the puzzles.

We report additional metrics recorded during evaluation such as Average Mistakes (Table 3) and Conversation Length (Table 4)

| Solver | Expert | Average Conversation Length (↓) | | | | | | | | | | |
|---|---|---|---|---|---|---|---|---|---|---|---|---|
| | | Button | Dog | Wire | Who | LED | Memory | Keypad | Password | Color | Maze | Overall |
| | GPT4V | 2.00 | 2.00 | 2.67 | 3.00 | 3.67 | 7.67 | 6.67 | 6.00 | 20.00 | 2.67 | 5.30 |
| Human | GPT4o | 2.67 | 2.00 | 3.00 | 6.00 | 1.33 | 8.33 | 10.50 | 3.00 | 20.00 | 2.67 | 5.79 |
| | InternVL8b | 4.00 | 7.00 | 3.00 | 9.50 | 18.50 | 20.00 | 19.50 | 13.00 | 20.00 | 16.00 | 12.38 |
| GPT4o | | 3.67 | 10.00 | 2.33 | 2.33 | 17.00 | 13.33 | 6.67 | 11.33 | 20.00 | 20.00 | 10.67 |
| GPT4V | Human | 8.33 | 5.67 | 6.67 | 2.33 | 15.67 | 15.33 | 2.00 | 18.00 | 15.50 | 20.00 | 10.79 |
| InternVL8b | | 5.50 | 4.00 | 5.67 | 3.33 | 14.00 | 17.00 | 7.67 | 11.00 | 20.00 | 20.00 | 10.93 |
| GPT4V | GPT4V | 8.20 | 5.40 | 2.20 | 3.40 | 15.10 | 15.80 | 12.90 | 15.60 | 20.00 | 18.00 | 11.66 |
| Random | InternVL | 5.17 | 4.21 | 3.80 | 4.48 | 12.96 | 20.00 | 19.88 | 20.00 | 20.00 | 19.79 | 12.98 |
| GPT4o | GPT4o | 9.70 | 8.90 | 2.20 | 6.90 | 20.00 | 19.00 | 14.70 | 20.00 | 20.00 | 20.00 | 14.14 |
| InternVL | GPT4o | 5.20 | 6.90 | 2.60 | 12.30 | 20.00 | 17.90 | 20.00 | 20.00 | 20.00 | 20.00 | 14.49 |
| QwenVL2b | QwenVL2b | 3.50 | 11.60 | 14.30 | 16.20 | 4.60 | 20.00 | 18.20 | 20.00 | 20.00 | 20.00 | 14.84 |
| InternVL26b | InternVL26b | 8.10 | 5.90 | 5.80 | 16.40 | 15.90 | 18.60 | 18.70 | 20.00 | 20.00 | 20.00 | 14.94 |
| InternVL8b | QwenVL7b | 4.30 | 4.70 | 16.40 | 17.00 | 15.60 | 20.00 | 18.40 | 20.00 | 20.00 | 20.00 | 15.64 |
| InternVL8b | InternVL8b | 5.50 | 6.30 | 14.70 | 13.20 | 20.00 | 18.80 | 20.00 | 20.00 | 20.00 | 19.90 | 15.84 |
| QwenVL7b | GPT4o | 9.80 | 9.00 | 10.45 | 15.90 | 20.00 | 18.10 | 16.80 | 20.00 | 20.00 | 20.00 | 15.95 |
| QwenVL7b | InternVL8b | 7.80 | 6.00 | 12.50 | 14.40 | 20.00 | 19.00 | 20.00 | 20.00 | 20.00 | 20.00 | 15.97 |
| QwenVL7b | QwenVL7b | 8.90 | 12.10 | 14.70 | 18.20 | 17.30 | 20.00 | 18.60 | 20.00 | 20.00 | 20.00 | 16.98 |

Table 4: Average conversation length for various puzzles. We average the length of agent dialogue over 10, 3, and 100 independent runs of each puzzle for the AI-AI, AI-Human, and random settings. The overall column is an average dialogue length across all the puzzles.

## D  AGENT PROMPTS

**Solver Prompt:**
```
You are the solver in a cooperative game involving
solving puzzles.  As the solver, you are presented
with an image of the puzzle, along with possible
actions you may take.  You should only attempt
some actions if you are certain of the solution.
Otherwise, you should describe the image and ask
the expert.  When asking the expert, keep in mind
the expert cannot see the image.  Your description
should be concise but also detailed enough to convey
the details to the expert through text only.  Once
you are certain of the solution, respond with just
the name of the action you chose.  If in a puzzle
you can take multiple steps to solve it, you could
output a list of action names, separated by the line
break \n and in the sequential order to be executed.
ONLY FINISH THE SOLVER'S DIALOGUE.
```

**Expert Prompt:**
```
You are the expert in a cooperative game involving
solving puzzles.  As the expert, you hold the puzzle
solution manual, containing vital information on
various modules and their corresponding solution
procedures.  Your task is to listen carefully to the
solver's descriptions of the puzzles and provide
clear and accurate instructions to guide them
through the solution.  Be as concise and precise in
your instructions as possible.  If the solver does
not provide you with enough information, ask for
clarification if needed.  ONLY FINISH THE EXPERT'S
DIALOGUE.
```

