# OpenReview forum: "COMMA: A Communicative Multimodal Multi-Agent Benchmark"
_ICLR.cc/2025/Conference — Submitted to ICLR 2025_

### Official Review · Reviewer_X6F7 · 2024-10-18

**Soundness:** 3
**Presentation:** 3
**Contribution:** 2
**Rating:** 3
**Confidence:** 3

**Summary:**

Benchmarks on both multimodal models and multi-agent benchmarks have been proposed. This paper studied the interaction of the two---how about multimodal multi-agent benchmarks? The authors hand-crafted 10 tasks that target 4 different capabilities. With evaluation over various models, a series of insights were gained.

**Strengths:**

+ Careful analysis of the experiment results provide insight into the current deficiencies of language models.

**Weaknesses:**

+ The proposed tasks are all toys. This causes the contribution of this paper to be limited even though it provides a reasonable starting point. As a contrasting example, in the original debate paper (Du et al.) the benchmarks that were used were specific capability benchmarks popular elsewhere.

**Questions:**

+ Is there a fundamental asymmetry between the expert and the solver? I'm asking because in the experiments, (a, b) and (b, a) are not always both tested, leading to a particularly long table 1 that is more difficult to interpret than necessary.

---

> ### Author Response · Authors · 2024-11-21
> **Response to Reviewer**
>
> We thank the reviewer for their time and helpful feedback!
>
> **Weakness: The proposed tasks are all toys**
>
> Although the tasks seem simple, we believe that the tasks in the benchmark test a variety of **cognitive abilities** of the agents. The use of simple tasks in our benchmark is inspired by established principles in psychology, where intelligence is widely regarded as the ability to learn from experience, adapt to the environment, and employ cognitive problem-solving skills. Many classical cognitive science papers have shown that even simple tests can be effective at measuring cognitive ability (1, 2), and standardized intelligence tests, such as MENSA and the Wechsler Intelligence Scale for Children, often use straightforward puzzles to assess these abilities. Despite their simplicity, these tasks reveal significant limitations in SOTA models such as GPT-4o, which perform poorly and struggle to outperform a random baseline.
>
> Additionally, the simplicity of the tasks allows us to modify configs and do quicker testing, as the main focus of our work is to provide a **composable framework** for communication between multimodal agents to assess their intelligence. Our framework can be customized by future users who wish to add their own more challenging puzzles and manuals to simulate their own more realistic scenarios. In our code release, you may find template files for adding your own agents and puzzles.
>
> 1.) [Development of cognitive control and executive functions from 4 to 13 years: Evidence from manipulations of memory, inhibition, and task switching](https://pmc.ncbi.nlm.nih.gov/articles/PMC1513793/)
>
> 2.) [Executive functions and achievements in school: Shifting, updating, inhibition, and working memory](https://pubmed.ncbi.nlm.nih.gov/16707360/)
>
> **Q1: Asymmetry between Solver and Expert**
>
> You are correct that there is an asymmetry between the expert and the solver and we do not always test (a,b), (b,a). Since our submission, we have tested more combinations and models. In our revised draft, we will reorganize the data in Table 1, potentially splitting the results into separate tables, to improve clarity and ease of interpretation.
>
> If we’ve successfully addressed your concerns, we’d greatly appreciate it if you could consider updating your score! Once again, we are incredibly grateful for your time and feedback!

---

> > ### Comment · Reviewer_X6F7 · 2024-11-22
> > **Comment**
> >
> > Thanks for the reply.
> >
> > For 1, this can't explain the lack of experiment on real datasets. Not sure this is more of a cogsci contribution paper.
> >
> > I will keep my ratings.

---

> > > ### Author Response · Authors · 2024-11-26
> > > **Response to Reviewer**
> > >
> > > Thank you for the response!
> > >
> > > We appreciate the concern about the lack of experiments of realistic data. However, we would like to clarify that our study intentionally avoids such scenarios to maintain a clear focus on analyzing the fundamental limitations of VLM agents. Tackling these edge cases often makes it challenging to disentangle different factors in the analysis, potentially shifting attention away from understanding core agentic capabilities. Besides, there are many impactful studies which analyze the reasoning capabilities of text-only LLMs using simple puzzles like Game of 24 (1) and Minesweeper (2). This recently published survey paper also highlights more of them and calls for similar benchmarks (3).
> > >
> > > Our primary goal is not to apply LLMs to a specific real-world task but to provide a structured and comprehensive analysis of their agentic abilities. By adopting this approach, we aim to offer foundational insights that can guide future advancements in agentic understanding and practical applications alike.
> > >
> > > 1.) [Tree of Thoughts: Deliberate Problem Solving with Large Language Models](https://arxiv.org/pdf/2305.10601)
> > >
> > > 2.) [Assessing Logical Puzzle Solving in Large Language Models: Insights from a Minesweeper Case Study ](https://aclanthology.org/2024.naacl-long.4.pdf)
> > >
> > > 3.) [Puzzle Solving using Reasoning of Large Language Models: A Survey](https://aclanthology.org/2024.emnlp-main.646.pdf)

---

### Official Review · Reviewer_t4YJ · 2024-10-19

**Soundness:** 2
**Presentation:** 2
**Contribution:** 3
**Rating:** 5
**Confidence:** 4

**Summary:**

This paper endeavors to assess the capability of multiple multimodal models to collaborate in accomplishing complex tasks when receiving different levels of information. The authors stipulate that this collaboration process should involve communication between the models through language, facilitating model-human cooperation. As a comprehensive benchmark, COMMA evaluates four types of capabilities during the collaborative task completion process. After a thorough assessment of both model-human and model-model collaborations, the authors discovered that even the most potent closed-source models performed inadequately in their tests.

**Strengths:**

As a comprehensive benchmark, the authors have evidently dedicated substantial effort. Regarding task distribution, the authors constructed ten subtasks that nearly comprehensively cover various aspects potentially implicated in collaborative task completion. From the perspective of model testing, the authors extensively evaluated numerous existing multimodal models, both closed-source and open-source, thereby effectively highlighting the limitations of current multimodal models. Finally, the authors classified the capabilities tested by their benchmark and analyzed common error scenarios, providing guidance for the improvement of these multimodal models.

**Weaknesses:**

The establishment of a benchmark is undeniably a demanding task. However, I wish to express a few concerns. Firstly, in this work, what is the distinction between an agent and a model? In other words, in my previous reviews, I have consistently employed the term "model" rather than "agent" as I believe that this paper is essentially evaluating the capabilities of models and not so-called agents. If you are evaluating the capabilities of agents, please provide a definition of an agent, especially in contrast to a model. If I wish to utilize your framework to evaluate the performance of well-known agents such as CAMEL AI, AutoGen, XAgent, and other open-source agents, how can I integrate these frameworks into your evaluation system? Currently, it appears to me that you are actually testing models rather than agents. It would help if you encapsulated the models within your agent framework, such as your solver and expert agents, prior to testing them rather than directly testing the capabilities of other agents.

Furthermore, I believe that the writing of this paper seems somewhat hasty. The figure in Figure 1 is not aesthetically pleasing. At the very least, you could consider using a monotone palette of black, white, and blue for the main figure to provide a clearer perspective for the reader. Another issue I noticed with the writing is that at line 115, there is an uncompiled citation that appears as a "?" symbol. While this is not a serious error, it does detract from the reader's experience. I suggest that the authors revise the writing, especially for a possible camera-ready submission.

Although I have raised these concerns, I also greatly appreciate your work. For benchmark papers, I typically take the time to review the data or even conduct a small test myself. I have noticed that you have not released the code. If possible, could you include some code or test cases during the rebuttal phase?

**Questions:**

See the weakness please.

---

> ### Author Response · Authors · 2024-11-21
> **Response to Reviewer**
>
> Thank you for your time and valuable feedback! We appreciate your insights and suggestions for improvement.
>
> **Distinction between model vs agent?**
>
> In this work, we use the term agent to describe a model which can interact dynamically with the environment, make decisions, and execute tasks autonomously. In contrast, we refer to a model as the underlying computational framework (e.g., a large language model like GPT) that generates predictions or outputs based on given inputs. Given that our framework leverages models like GPT-4o and InternVL as backbone components, encapsulating them within an agent-like framework to solve tasks, we acknowledge that while we are evaluating the underlying models, we do so by treating them as agents. We will be sure to clarify this distinction in our draft.
>
> CAMEL AI, AutoGen, and XAgent similarly encapsulate various LLM models but instead focus on building and optimizing autonomous agents for specific workflows or general-purpose tasks. However, applying such frameworks directly within our evaluation as if they were single models presents significant challenges. Nevertheless, our code supports adding any framework which can accept multimodal (vision + language) input and generate natural language output.
>
>
> **Hasty Writing and Figure Improvement**
>
> Regarding your comment about the writing feeling hasty, would it be possible for you to provide more specific feedback on which sections or aspects you found to be rushed? This would greatly help us in making more targeted revisions.
>
> We are also actively working on enhancing the aesthetics of Figure 1 and 2, and we have updated the figure to use a monotone palette to improve clarity. Additionally, we have fixed the uncompiled citation and will share the new draft by the end of the discussion period.
>
> **Code Release**
>
> While we are still working on a final code release, we are happy to release a demo version for your experimentation (see supplementary material). In the files, you can also see some conversations between the agents during our testing in a folder called “outputs”. There are some setup instructions in the README, but we recommend the following:
>
> 1.) Download the .zip file, extract its contents, and navigate to the folder with a command line
>
> 2.) Create a new anaconda environment, activate it, then run **pip install -r requirements.txt**
>
> 3.) Fill in **config/keys.json** with the API keys for the deployments you may have to any of those models.
>
> 4.) Fill in **config/experiment_config.json** to customize what kind of agent is the solver/expert. (We recommend leaving it as is, and trying out human solver + GPT-4o expert to make it interactive)
>
> 5.) Run **python main.py –config config/puzzle.json –model_config experiment_config.json –gui True**
>
> We hope this can give you an opportunity to try out our framework and we are happy to answer any questions or concerns you may have! If we’ve successfully addressed your concerns, we’d greatly appreciate it if you could consider updating your score! Once again, we are incredibly grateful for your time and feedback!

---

> > ### Comment · Reviewer_t4YJ · 2024-11-24
> > **Fix the typo Immediately**
> >
> > Sorry for pointing this out once again. But see your line 115:
> >
> > et al. (2023); Hong et al. (2024;?), scientific discovery simulation Wuet al. (2023), and social
> >
> > There is an "?" in the second citation. I feel it's not appropriate for me to stress this twice.

---

> > > ### Author Response · Authors · 2024-11-24
> > > **Typo Fixed**
> > >
> > > Thank you once again for pointing this out. While we were planning on sharing a new draft at the end of the rebuttal period, we just have just uploaded a draft now where the typo has been fixed.

---

### Official Review · Reviewer_FrT1 · 2024-10-24

**Soundness:** 3
**Presentation:** 3
**Contribution:** 2
**Rating:** 5
**Confidence:** 4

**Summary:**

The paper presents COMMA, a benchmark to assess the two-agent collaborative ability of VLMs. The benchmark consists of 10 simulators that provide an environment for a Solver and an Expert agents. The simulators are designed in a way that each of the two agents is not able to arrive at the success state on its own. The authors show that modern VLMs acting as both a Solver and an Expert are not able to surpass the random baseline. On the contrary, a Human in the loop acting as one of the two agents in collaboration with best VLMs are able to surpass the random baseline with varied success.

**Strengths:**

[1] A benchmark for assessment of the collaborative abilities of VLMs is very valuable. Moreover, the presented benchmark assesses VLMs, the most functional single models to date, as opposed to non multimodal text-only LLMs.

[2] The analysis of the failure cases along with figures 3 and 4 is very insightful.

[3] The choice of models to evaluate is good: QuenVL and InternVL are at the top of ​​OpenVLM Leaderboard.

**Weaknesses:**

[1] According to Figure 5 Left, the random baseline is very strong meaning that the benchmark is not well designed. This mostly is attributed to the small number of choices and not penalizing wrong choices enough. The claim is that modern agents are not better than the random baseline is strong only if the random baseline is weak which is not the case. In general, intuitively, with random actions during “bomb defusal” I would not expect more than 5% or even 1% success rate for it to be a proper skill test.

[2] Button and Dog are not discriminative at 100% success rate almost for all combinations of agents.

[3] Results in Table 1 need standard deviation numbers (+/- one sigma).

[4] The paper lacks the Human-Human baseline.

[5] Overall the benchmark looks quite simplistic and is very likely to be mastered by the next generation of VLMs. The benchmark lacks assessment of more sophisticated VLM assessment like reasoning about relative positions and sizes of the objects.

[6] The benchmark cannot assess LLMs since it is tailored for VLMs.

[7] The benchmark does not evaluate VLMs in more than a 2-instance setting since all games are 2-instance.

**Questions:**

[1] How does your benchmark stop the expert from handing over the instruction to the solver?

[2] Why aren’t LLaVA, Llama 3.2 and Clause 3.5 evaluated? What about QuenVL 72B and InternVL 76 and 40B?

[3] A lot of citations are arxiv, even though they are the papers have been already published. Please update all the citations.

[4] A good example of how to make a random baseline stronger is MMLU-Pro which extends the original 4-choice MMLU to 10 choices to rebuke the random chance from 25% to 10%.

---

> ### Author Response · Authors · 2024-11-21
> **Response to Reviewer**
>
> We thank the reviewer for their time and helpful feedback!
>
> **W1: The random baseline is very strong meaning that the benchmark is not well designed**
>
> A1: We appreciate the reviewer highlighting this concern. We conducted additional experiments to further evaluate the robustness of the benchmark. Specifically, we ran the random agent twice: first by adding 5 dummy options to each puzzle and then by adding 10 dummy options. Results showed that the overall partial success rate decreased significantly to 36% (compared to the 56% reported in the paper).
>
> We believe the reason the random baseline performs relatively well in the original setting is due to the 20-turn limit, which allows even a random agent to exhaustively attempt many puzzles. However, we emphasize that our benchmark evaluates not only success rate but also complementary metrics such as average conversation length, errors made, and nuanced failure modes (e.g., imitation, misinterpretation). These metrics distinguish our approach from static benchmarks like MMLU-Pro, where only a single attempt is allowed. Additionally, these results demonstrate the benchmark's resilience under modifications, suggesting its robustness to changes.
>
> **W2: Button and Dog are not discriminative at 100% success rate almost for all combinations of agents.**
>
> A2: We acknowledge the simplicity of the Button and Dog puzzles. However, these were intentionally designed as part of a spectrum of puzzle difficulties to evaluate a broad range of capabilities. While these puzzles are easier to solve, we observe some variation in performance between configurations, such as Human-GPT4V and GPT4V-GPT4V, indicating that these puzzles still provide valuable insights into agent collaboration dynamics. Moreover, simpler puzzles like these serve a purpose by anchoring the evaluation and allowing us to assess basic capabilities. Future iterations of the benchmark could replace these puzzles with more complex alternatives, but we believe their inclusion currently offers utility in evaluating both strong and weaker models.
>
> **W3: Results in Table 1 need standard deviation numbers (+/- one sigma).**
>
> A3: Thank you for pointing this out, we will be sure to include it after the review period.
>
> **W4: The paper lacks the Human-Human baseline.**
>
> A4: We decided not to include a Human-Human baseline because all puzzles are solvable following the instruction manual, with clear rules dictating the next move at each state. We expect individuals with reasonable intelligence to achieve near-perfect scores.
>
> **W5: Overall the benchmark looks quite simplistic and is very likely to be mastered by the next generation of VLMs..**
>
> Although the tasks seem simple, we believe that the tasks in the benchmark test a variety of cognitive abilities of the agents. The use of simple tasks in our benchmark is inspired by established principles in psychology, where intelligence is widely regarded as the ability to learn from experience, adapt to the environment, and employ cognitive problem-solving skills. Many classical cognitive science papers have shown that even simple tests can be effective at measuring cognitive ability (1, 2), and standardized intelligence tests, such as MENSA and the Wechsler Intelligence Scale for Children, often use straightforward puzzles to assess these abilities. Despite their simplicity, these tasks reveal significant limitations in SOTA models such as GPT-4o, which perform poorly and struggle to outperform a random baseline.
>
> Additionally, the simplicity of the tasks allows us to modify configs and do quicker testing, as the main focus of our work is to provide a framework for communication between multimodal agents to assess their intelligence. Our framework can be customized by future users who wish to add their own more challenging puzzles and manuals to simulate their own more realistic scenarios. In our code release, you may find template files for adding your own agents and puzzles.
>
> 1.) [Development of cognitive control and executive functions from 4 to 13 years: Evidence from manipulations of memory, inhibition, and task switching](https://pmc.ncbi.nlm.nih.gov/articles/PMC1513793/)
>
> 2.) [Executive functions and achievements in school: Shifting, updating, inhibition, and working memory](https://pubmed.ncbi.nlm.nih.gov/16707360/)
>
> **W6: The benchmark cannot assess LLMs since it is tailored for VLMs.**
>
> A6: We acknowledge that our benchmark is designed primarily for VLMs. However, we conducted tests with LLMs like GPT-4 (text-only mode) by providing detailed textual descriptions of puzzle states as input. These experiments demonstrated that the benchmark framework can be adapted for LLMs when multimodal inputs are unavailable. That said, the primary goal of this work is to advance the evaluation of multimodal agents specifically.

---

> > ### Author Response · Authors · 2024-11-21
> > **Response to Reviewer (Continued)**
> >
> > **W6: The benchmark cannot assess LLMs since it is tailored for VLMs.**
> >
> > A6: We acknowledge that our benchmark is designed primarily for VLMs. However, we conducted tests with LLMs like GPT-4 (text-only mode) by providing detailed textual descriptions of board states as input. These experiments demonstrated that the benchmark framework can be adapted for LLMs when multimodal inputs are unavailable. That said, the primary goal of this work is to advance the evaluation of multimodal agents specifically. Given the nascency of VLMs, we believe it is crucial to develop tools that are tailored to their unique capabilities, while ensuring that the framework can accommodate broader use cases, including text-only models.
> >
> >
> > **W7: The benchmark does not evaluate VLMs in more than a 2-instance setting since all games are 2-instance.**
> >
> > A7: Our current design focuses on pairwise collaborative communication to thoroughly assess core interaction and reasoning capabilities. Introducing more agents would increase task complexity and potentially obfuscate key evaluation metrics. However, we agree that scaling to multi-agent scenarios is an important future direction. Our framework is inherently flexible, allowing for straightforward extensions to multi-agent settings (e.g., multiple solvers with a single expert or multiple experts). We consider this an exciting avenue for future exploration, leveraging the groundwork laid by this benchmark.
> >
> > **Q1: How does your benchmark stop the expert from handing over the instruction to the solver?**
> >
> > A1: Our benchmark does not impose a strict restriction to prevent the expert from handing over the instructions verbatim. However, the prompt encourages the expert to communicate in a succinct and conversational manner, focusing on collaboration rather than simply copying instructions.
> > To address this concern further, we have analyzed word count as an additional metric since our submission to determine if the expert is adhering to a conversational style. This metric can be included in our final submission to provide a more detailed analysis of the expert’s behavior. Future iterations of the benchmark could incorporate additional constraints if necessary to further discourage direct instruction handovers.
> >
> >
> > **Q2. Why aren’t LLaVA, Llama 3.2 and Clause 3.5 evaluated? What about QuenVL 72B and InternVL 76 and 40B?**
> >
> > A2: We attempted to evaluate LLaVA on our benchmark but faced challenges due to the distribution shift between the VQA tasks it was trained on and the collaborative nature of our benchmark. Despite our efforts, LLaVA failed to perform meaningfully, which is why it was excluded from the results. Since the initial submission, we have incorporated Llama 3.2 as an agent and will include its results in the final version. Regarding Claude 3.5, we did not have access to its API at the time of testing, and for QwenVL and InternVL, we opted for their smaller versions due to limited compute resources. We acknowledge the importance of including state-of-the-art models and will aim to expand our evaluation in future work as resources become available.
> >
> >
> > **Q3. A lot of citations are arxiv, even though they are the papers have been already published. Please update all the citations.**
> >
> > A3: Thank you for pointing this out! We will ensure that all citations in the final version are updated to reflect the official venues where the papers were published.
> >
> > **Q4. A good example of how to make a random baseline stronger is MMLU-Pro which extends the original 4-choice MMLU to 10 choices to rebuke the random chance from 25% to 10%.**
> >
> > A4: See response to weakness 1. Additionally, here are the results from our runs.
> >
> > | Metric (Additional dummy options) | Button | Dog | SimpleWire | Who | LED | Memory | KeyPad | Password | Colour | Maze | Overall |
> > | -------- | ------- | ------- | ------- | ------- | ------- | ------- | ------- | ------- | ------- | ------- | ------- |
> > | Partial Score Results (5) | 0.40 | 1.00 | 0.75 | 0.80 | 0.70 | 0.24 | 0.30 | 0.00 | 0.2 | 0.11 | 0.46 |
> > | Partial Score Results (10)| 0.00   | 0.90 | 0.80       | 0.70 | 0.41 | 0.16   | 0.25   | 0.00     | 0.12   | 0.38 | 0.37    |
> > | Success Rate Results (5) | 0.40   | 1.00 | 0.75       | 0.80 | 0.50 | 0.00   | 0.10   | 0.00     | 0.00   | 0.00 | 0.37    |
> > | Success Rate Results (10)| 0.00   | 0.90 | 0.80       | 0.70 | 0.20 | 0.00   | 0.00   | 0.00     | 0.00   | 0.00 | 0.26    |
> > | Conversation Length (5)  | 15.60  | 9.30 | 5.45       | 7.80 | 15.20| 20.00  | 19.50  | 20.00    | 20.00  | 20.00| 15.19   |
> > | Conversation Length (10) | 20.00  | 9.60 | 7.80       | 9.50 | 17.70| 20.00  | 20.00  | 20.00    | 20.00  | 20.00| 16.46   |
> > | Word Count Results (5)   | 196.00 |132.40| 71.45      | 94.00|197.40| 242.00 | 235.00 | 242.00   | 242.00 |242.00| 188.26  |
> > | Word Count Results (10)  | 242.00 |118.20| 116.20     |121.40|217.60| 242.00 | 242.00 | 242.00   | 242.00 |242.00| 202.54  |

---

> > > ### Comment · Reviewer_FrT1 · 2024-11-22
> > >
> > > Thank you for your answers and extra experiments to analyze the random baseline. I intend to keep my score. However, I strongly believe that the contribution should be reinforced with the following:
> > > 1. Reworked in a way that has a random baseline that is not higher than 10% for all metrics.
> > > 2. Extended with more tasks that are less toy-like since the authors claim that their framework allows it to be done easily.
> > > 3. High-score tasks should be removed as non-discriminative.
> > > 4. It is important for the human baseline to be measured experimentally, including design choices like a possible time limit.
> > > 5. A more convincing explanation is needed to justify why the seemingly toy problems are a strong test of communicative abilities.
> > > 6. Handling over the instruction should be detected and either moderated according to a defined condition or measured as an extra metric.
> > > 7. 1-sigma confidence intervals should be plotted/reported (I still do not see them in the answers).

---

### Official Review · Reviewer_iCwh · 2024-11-03

**Soundness:** 2
**Presentation:** 3
**Contribution:** 2
**Rating:** 5
**Confidence:** 3

**Summary:**

This paper proposes a novel benchmark to study multimodal agents that communicate with humans to solve tasks. The authors select ten different puzzles that incorporate multimodal, memory-dependent, multi-step, and real-time elements. They evaluate four different vision-language models (VLMs) across these tasks in both Human-AI and AI-AI settings. From their analysis, the authors identify four common issues for VLMs within this benchmark: miscommunication, role misunderstanding, repetition, and misinterpretation.

**Strengths:**

The focus on Human-AI natural language communication to solve multimodal tasks addresses a valuable research problem, and this paper establishes a promising starting point of this direction using puzzles. The authors provide strong qualitative analysis, offering insights into the weaknesses of various VLMs as agent backbones. These examples help reveal common challenges among VLMs and highlight differences between them, contributing useful knowledge for future research.

**Weaknesses:**

In the AI-human experiments, only three data points are used for each puzzle, which is insufficient, given the relatively small performance differences between models. Additionally, the tasks are not exciting enough because solving puzzles is somewhat far from real-world scenarios, so this benchmark may serve primarily as an introductory step in studying this field.

**Questions:**

1. Could you increase the number of runs per puzzle? For AI-AI and AI-human settings, the sample sizes of 10 and 3, respectively, seem limited for evaluating a VLM on puzzle tasks.

2. The example in the bottom right corner of Figure 2 doesn’t clearly illustrate “Misinterpretation”—miscounting the number of dogs doesn’t seem to fall under this category.

3. Could you provide human-human results for reference? Including success rates and conversation lengths could offer a useful upper bound for current performance.

4. Could you compare human behavior to AI on certain challenging puzzles, such as the password puzzle? It’s unclear why most AI-AI models completely fail on this puzzle.

5. GPT-4V+human succeeded in some maze puzzles where GPT-4o+human failed completely, which makes me confused. Could you explain the reason behind it?

---

> ### Author Response · Authors · 2024-11-21
> **Response to Reviewer**
>
> We thank the reviewer for their time and helpful feedback!
>
> **The tasks aren’t exciting enough (Weakness)**
>
> Although the tasks seem simple, we believe that the tasks in the benchmark test a variety of **cognitive abilities** of the agents. The use of simple tasks in our benchmark is inspired by established principles in psychology, where intelligence is widely regarded as the ability to learn from experience, adapt to the environment, and employ cognitive problem-solving skills. Many classical cognitive science papers have shown that even simple tests can be effective at measuring cognitive ability (1, 2), and standardized intelligence tests, such as MENSA and the Wechsler Intelligence Scale for Children, often use straightforward puzzles to assess these abilities. Despite their simplicity, these tasks reveal significant limitations in SOTA models such as GPT-4o, which perform poorly and struggle to outperform a random baseline.
>
> Additionally, the simplicity of the tasks allows us to modify configs and do quicker testing, as the main focus of our work is to provide a **composable framework** for communication between multimodal agents to assess their intelligence. Our framework can be customized by future users who wish to add their own more challenging puzzles and manuals to simulate their own more realistic scenarios. In our code release, you may find template files for adding your own agents and puzzles.
>
> 1.) [Development of cognitive control and executive functions from 4 to 13 years: Evidence from manipulations of memory, inhibition, and task switching](https://pmc.ncbi.nlm.nih.gov/articles/PMC1513793/)
>
> 2.) [Executive functions and achievements in school: Shifting, updating, inhibition, and working memory](https://pubmed.ncbi.nlm.nih.gov/16707360/)
>
> **Q1: The sample sizes per puzzle for AI-AI and AI-human settings are insufficient. (Weakness+Q1)**
>
> A1: We appreciate your concern regarding the number of data points and the nature of the tasks. However, when considering the dynamics of human-AI interaction, each puzzle inherently requires multiple rounds of dialogue between the user and the model, resulting in considerable time expenditure. The study involved 30 experiments across 10 puzzles in human-AI setting and 100 experiments in AI-AI setting for each pair of VLMs, showing they still have gaps in collaborative reasoning abilities from human performance.
>
> In the future version, we will employ bootstrapping techniques to enhance the accuracy and robustness of the results we report. Thus, while the puzzles serve as controlled scenarios, the insights gained are significant, highlighting the current limitations and potential of VLMs in real-world collaborative contexts.
>
> **Q2: Figure 2 doesn’t illustrate “Misinterpretation”—miscounting the number of dogs doesn’t seem to fall under this category.**
>
> A2: In the context of Figure 2, the miscounting of the number of dogs is an instance where the solver's misunderstanding of the visual information directly results in incorrect instructions from the expert. This illustrates how a misinterpretation of the present data can lead to a failure in task execution. We are open to refining the example or incorporating additional explanations to enhance clarity and better convey the essence of "misinterpretation" in this context.
>
> **Q3: Could you provide human-human results for reference?**
>
> A3: We did not include and report human-human setting because all puzzles are solvable for human following the instruction manual, with clear rules dictating the next move at each state. We expect individuals with reasonable intelligence to achieve near-perfect scores.
>
> **Q4: Could you compare human behavior to AI on certain challenging puzzles, such as the password puzzle? Why do most AI-AI models completely fail on this puzzle?**
>
> A4: The primary challenge AI models face with the password puzzle is the limitation of only 20 conversation turns. This constraint requires predicting long action sequences in a single attempt, which is inherently difficult. Unlike humans, who can intuitively adjust and refine their strategies with more feedback, AI struggles to perform complex reasoning and adapt within these limits. This underscores the need for more flexible interaction frameworks to improve AI problem-solving in such scenarios.
>
> **Q5: GPT-4V+human succeeded in some maze puzzles where GPT-4o+human failed completely.**
>
> A5: It is important to note that GPT-4V+human succeeded in just one more maze puzzle than GPT-4o+human. This minor difference can be attributed to variations in human interaction and model capabilities such as how visual information is interpreted and integrated. So the slight discrepancies are normal in collaborative problem-solving.
>
> If we’ve successfully addressed your concerns, we’d greatly appreciate it if you could consider updating your score! Once again, we are incredibly grateful for your time and feedback!

---

> > ### Comment · Reviewer_iCwh · 2024-11-22
> > **Official Comment by Reviewer iCwh**
> >
> > Thank you for the response.
> >
> > *Response to the first paragraph:*
> >
> > I acknowledge that puzzles are a good way to measure cognitive ability. However, the paper has not yet adequately explained this intuition from a cognitive perspective.
> >
> >
> > *Response to A1:*
> >
> > I understand your challenges. My concern is that, with a limited number of data points, the differences between models might be caused by just one or two datapoints, making the overall score differences potentially meaningless. In other way, if you can demonstrate the significance of these differences, it would also address this concern. Additionally, if this benchmark requires such intensive human evaluation, it would be difficult for others to follow or replicate this work on new VLMs.

---

> > > ### Author Response · Authors · 2024-11-24
> > > **Response to Reviewer**
> > >
> > > Thank you for your response! We have added some intuition for why we use simple puzzles from a cognitive perspective in Section 3. We are also actively working on adding template files for future users to add their own agents and puzzles, and a preliminary demo version of our code is available in the supplementary materials.
> > >
> > > We acknowledge the concern regarding the potential influence of a limited number of data points on the observed differences between models. To address this, we have undertaken additional efforts since our initial submission to extract more meaningful information from the data that we have collected.
> > >
> > > Specifically, we have worked on automating the evaluation process using GPT-4o, focusing on the key error categories identified in our study: misinterpretation, miscommunication, roleplay confusion, and repetition loops. We first use GPT-4o to evaluate a conversation for each individual puzzle run i.e. using it to identify which of the errors are present in the conversation and assigning a binary score. We then group the data by the unique solver-expert pairings and average across all puzzle runs for that pairing to produce the table shown below, showing the percentage of cases where such an error was observed. This automated evaluation allowed us to systematically assess the significance of the observed differences across models and mitigate concerns about the influence of outliers or small sample sizes.
> > >
> > > We have included these new results below to provide a clearer and more rigorous demonstration of the significance of the differences. Furthermore, this automated approach reduces reliance on intensive human evaluation, enhancing the reproducibility and scalability of our methodology for future benchmarking on new VLMs. We find that our analysis reveals significant variation in error rates for different settings. For instance, we find that a repetition loop is the most common error that occurs (64% on average) and, within that error category, there is a range of 5% to 88% of the instances having such an error. We feel this showcases our framework's ability to capture more nuanced errors to differentiate between model settings.
> > > We hope these additional efforts address the reviewer's concerns and strengthen the contribution.
> > >
> > >
> > > | Solver-Agent Pair        | Mean Roleplay Confusion | Mean Misinterpretation | Mean Repetition Loop | Mean Miscommunication |
> > > |--------------------------|-------------------------|-------------------------|-----------------------|-----------------------|
> > > | GPT4VAgent-GPT4VAgent    | 0.08             | 0.28              | 0.48         | 0.48             |
> > > | GPT4VAgent-HumanAgent    | 0.10            | 0.28               | 0.45          | 0.45             |
> > > | GPT4oAgent-GPT4oAgent    | 0.06             | 0.36              | 0.69             | 0.49             |
> > > | GPT4oAgent-HumanAgent    | 0.07               | 0.13           | 0.40                  | 0.50              |
> > > | GPT4oAgent-QwenVL7bAgent | 0.09                   | 0.31                   | 0.78                 | 0.49                 |
> > > | HumanAgent-GPT4VAgent    | 0                      | 0.14               | 0.06            | 0.20                  |
> > > | HumanAgent-GPT4oAgent    | 0.03        | 0.30               | 0.12            | 0.33           |
> > > | HumanAgent-InternVLAgent | 0.05               | 0.29              | 0.76            | 0.52           |
> > > | InternVL26bAgent-InternVL26bAgent | 0.05          | 0.24                   | 0.81                 | 0.44                 |
> > > | InternVL7bAgent-InternVL7bAgent   | 0.22       | 0.16           | 0.82           | 0.38          |
> > > | InternVL8bAgent-QwenVL7bAgent     | 0.13           | 0.11                   | 0.82                 | 0.37                 |
> > > | InternVLAgent-GPT4oAgent | 0.07           | 0.26             | 0.73       | 0.57      |
> > > | InternVLAgent-HumanAgent | 0.11            | 0.41             | 0.63          | 0.44          |
> > > | QwenVL2bAgent-QwenVL2bAgent       | 0.07           | 0.07                   | 0.78                 | 0.48                 |
> > > | QwenVL7bAgent-GPT4oAgent | 0.05            | 0.27             | 0.84          | 0.58        |
> > > | QwenVL7bAgent-InternVL8bAgent    | 0.04            | 0.10           | 0.88                 | 0.28                 |
> > > | QwenVL7bAgent-QwenVL7bAgent      | 0.06            | 0.16                   | 0.87                 | 0.45                 |
> > > | **Average**              | **0.07**           | **0.23**           | **0.64**         | **0.44**         |

---

> > > > ### Comment · Reviewer_iCwh · 2024-11-24
> > > >
> > > > I think this is a nice effort, as it enhances the benchmark by providing more informative results and aligns with the promising direction of automating experiments.
> > > >
> > > > However, my previous concerns still exist: while you present mean values in the table, it is difficult to determine whether the differences between these means are sound, especially when the number of datapoints is limited. I noticed another reviewer (FrT1) mentioned +/- sigma, which can also help with this concern.
> > > >
> > > > I believe a well-designed benchmark should enable researchers to conduct experiments on new VLMs within a reasonable time while ensuring that the results clearly demonstrate significant differences between models.

---

> ### Author Response · Authors · 2024-11-26
> **Response to Reviewer**
>
> We thank you again for your feedback on this matter. We understand your concern regarding the difficulty in determining whether the differences between mean values are statistically significant, especially with a limited number of data points. To address this, we have improved our GPT-4o judge by expanding the scoring range from a binary 0/1 to a larger 0-5 scale. This allows us to capture the severity of errors more precisely, ranging from minor issues affecting a single line to significant problems impacting the entire conversation between the expert and solver. Additionally, we now include the standard deviations (± sigma) alongside the mean values in our results. This provides a clearer picture of the variability in the data, enabling a better assessment of the statistical significance of the observed differences. We think an interesting highlight is how self-interactions among certain agents (InternVL7bAgent and QwenVL7bAgent) lead to significantly higher repetition loop errors compared to when humans are involved in the process (with them or with other agents), suggesting challenges in generating novel responses without human input.
>
> We have also added detailed templates to add new agents and puzzles in the supplementary materials, which we hope would make it much easier for anyone to extend our benchmarks for new multimodal settings and/or VLMs.
>
> | Solver             | Expert              | Roleplay Confusion | Misinterpretation | Repetition Loop | Miscommunication |
> |--------------------|---------------------|--------------------|-------------------|-----------------|------------------|
> | GPT4VAgent         | GPT4VAgent          | 0.03 ± 0.22        | 1.22 ± 1.30       | 1.83 ± 2.04     | 1.74 ± 1.36      |
> | GPT4VAgent         | HumanAgent          | 0.07 ± 0.26        | 0.90 ± 1.18       | 1.45 ± 1.64     | 1.55 ± 1.45      |
> | GPT4oAgent         | GPT4oAgent          | 0.11 ± 0.37        | 1.32 ± 1.45       | 2.75 ± 2.04     | 1.94 ± 1.50      |
> | GPT4oAgent         | HumanAgent          | 0.00 ± 0.00        | 0.63 ± 0.89       | 1.57 ± 1.91     | 1.67 ± 1.42      |
> | GPT4oAgent         | QwenVL7bAgent       | 0.17 ± 0.47        | 0.88 ± 1.10       | 3.72 ± 1.89     | 1.89 ± 1.48      |
> | HumanAgent         | GPT4VAgent          | 0.00 ± 0.00        | 0.63 ± 1.06       | 0.29 ± 0.96     | 1.00 ± 1.48      |
> | HumanAgent         | GPT4oAgent          | 0.00 ± 0.00        | 0.91 ± 1.21       | 0.39 ± 1.12     | 1.00 ± 1.48      |
> | HumanAgent         | InternVLAgent       | 0.24 ± 1.09        | 1.10 ± 1.34       | 3.14 ± 1.98     | 2.33 ± 1.56      |
> | InternVL26bAgent   | InternVL26bAgent    | 0.00 ± 0.00        | 0.89 ± 1.09       | 3.80 ± 1.96     | 2.16 ± 1.43      |
> | InternVL7bAgent    | InternVL7bAgent     | 0.27 ± 0.85        | 0.63 ± 1.06       | 4.22 ± 1.72     | 1.73 ± 1.61      |
> | InternVL8bAgent    | QwenVL7bAgent       | 0.04 ± 0.24        | 0.52 ± 0.98       | 3.73 ± 1.97     | 1.65 ± 1.53      |
> | InternVLAgent      | GPT4oAgent          | 0.01 ± 0.11        | 0.73 ± 1.13       | 3.49 ± 2.13     | 2.18 ± 1.68      |
> | InternVLAgent      | HumanAgent          | 0.00 ± 0.00        | 0.63 ± 1.15       | 2.59 ± 2.29     | 2.04 ± 1.63      |
> | QwenVL2bAgent      | QwenVL2bAgent       | 0.00 ± 0.00        | 0.24 ± 0.64       | 3.90 ± 1.91     | 1.91 ± 1.68      |
> | QwenVL7bAgent      | GPT4oAgent          | 0.02 ± 0.14        | 0.83 ± 1.25       | 3.51 ± 2.01     | 2.19 ± 1.60      |
> | QwenVL7bAgent      | InternVL8bAgent     | 0.05 ± 0.22        | 0.35 ± 0.76       | 4.28 ± 1.60     | 1.25 ± 1.43      |
> | QwenVL7bAgent      | QwenVL7bAgent       | 0.00 ± 0.00        | 0.51 ± 0.92       | 4.28 ± 1.58     | 2.40 ± 1.53      |

---

### Meta-Review · Area_Chair_Sa67 · 2024-12-19

**Metareview:**

This paper introduces a new multi-agent collaboration benchmark, COMMA, in which a solver agent and an expert agent must communicate and collaborate with each other to succeed. The results showed that VLMs alone cannot solve the tasks. On the other hand, human-AI collaboration exhibited a better performance. The findings highlighted the limits of current VLMs. The multimodal aspect of the benchmark is also particularly valuable. However, there are two critical concerns about the current benchmark that I think the revision and the rebuttal have not fully addressed:

1. The tasks are not realistic enough.

2. A strong random baseline may suggest important flaws in the benchmark design.

Given this, I would recommend further improvement before publication.

**Additional Comments On Reviewer Discussion:**

Reviewers reached a consensus after discussion. While several concerns such as sample sizes and variance, and the difference between a model and an agent have been addressed, there are still remaining critical concerns. In particular, reviewer FrT1 decreased their rating after realizing the flaws that may be suggested by the strong random baseline.

---

### Decision · Program_Chairs · 2025-01-22

Reject